# Deep Learning and Artificial Neural Networks for Spacecraft Dynamics, Navigation and Control

Stefano Silvestrini * and Michèle Lavagna

Department of Aerospace Science and Technologies, Politecnico di Milano, 20156 Milan, Italy
* Correspondence: stefano.silvestrini@polimi.it

**Abstract:** The growing interest in Artificial Intelligence is pervading several domains of technology and robotics research. Only recently has the space community started to investigate deep learning methods and artificial neural networks for space systems. This paper aims at introducing the most relevant characteristics of these topics for spacecraft dynamics control, guidance and navigation. The most common artificial neural network architectures and the associated training methods are examined, trying to highlight the advantages and disadvantages of their employment for specific problems. In particular, the applications of artificial neural networks to system identification, control synthesis and optical navigation are reviewed and compared using quantitative and qualitative metrics. This overview presents the end-to-end deep learning frameworks for spacecraft guidance, navigation and control together with the hybrid methods in which the neural techniques are coupled with traditional algorithms to enhance their performance levels.

**Keywords:** ANN; spacecraft; GNC; deep learning; dynamics; autonomous; control; navigation

## 1. Introduction

One of the major breakthrough in the last decade in autonomous systems has been the development of an older concept named Artificial Intelligence (AI). This term is vast and addresses several fields of research. Moreover, Artificial Intelligence is a broad term that is often confused with one of its sub-clustering terms. The well-known artificial neural networks (ANNs) are nearly as old as Artificial Intelligence, and they represent a tool, or a model, rather than a method by which to implement AI in autonomous systems. Nearly all deep learning algorithms can be described as particular instances of a standard architecture: the idea is to combine a dataset for specification, a cost function, an optimization procedure and a model, as reported in [1]. Actually, for guidance, navigation and control, using a dataset produces poor results due to distribution mismatching. Even for the case where training is done in a simulated environment but not during deployment, the need to update the dataset using simulated observations and actions during training is justified by the mentioned dataset distribution mismatch, as thoroughly presented in [2]. Additionally, updating the dataset with incremental observations tends to reduce overfitting problems. This survey presents the theoretical basis for the foundational work of [1,3–5]. In this overview, the focus is to catch a glimpse of the current trends in the implementation of AI-based techniques in space applications, in particular for what concerns hybrid applications of artificial neural networks and classical algorithms within the domains of guidance, navigation and control. Even though the survey is restricted to these domains, the topic is still very broad, and different perspectives can be found in recent surveys [6–12]. Most of the analyzed surveys focus on a limited application, deeply investigating the technical solutions for a particular scenario. Table 1 compares the existing works with this manuscript.

**Table 1.** Comparison between recent survey works and this review.

| Ref. | Highlights | This Review |
|---|---|---|
| [6] | The survey focuses on machine learning techniques in spacecraft control design. | This paper extends the review to navigation and estimation in space. |
| [7] | The survey is limited to the relative navigation task using deep learning | This paper extends the review to online estimation, AI-aided filtering and machine learning spacecraft control. |
| [8] | The survey thoroughly reviews multiple applications of machine learning techniques, particularly focusing on FDIR. Moreover, it reports a review of the most common Edge AI boards applicable to space-based systems. | This paper focuses on GNC applications, yielding also a mathematical tutorial for the development of some of the presented applications. |
| [9] | The survey thoroughly reviews end-to-end guidance and control applications based on AI. | This paper entails a significant discussion on the hybrid techniques that incorporate traditional algorithms to AI-based approaches. |
| [10] | The survey thoroughly reviews deep learning methods for unmanned aerial vehicles. | This paper focuses on GNC and estimation for space-based systems. |
| [12] | The survey focuses on reinforcement learning applications for spacecraft control | This paper extends the discussion to spacecraft navigation and estimation, together with tutorial-like analysis of common artificial neural network architectures. |

The range of applications is from preliminary spacecraft design to mission operations, with an emphasis on guidance and control algorithms coupled with navigation; finally, perturbed dynamics reconstruction and classification of astronomical objects are emerging topics. Due to the very large number of applications, it is the authors' intent to narrow down the discussion to spacecraft guidance, navigation and control (GNC), and the dynamics reconstruction domain. Nevertheless, besides those falling into the above-mentioned domains, the most promising applications in space of AI-based techniques are mentioned within the discussion of the most common network architectures.

The major contributions of this paper are:

- to introduce the bases of machine learning and deep learning that are rapidly growing within the space community;
- to present a review of the most common artificial neural network architectures used in the space domain, together with emerging techniques that are still theoretical;
- to present specific applications extrapolating the underlying cores of the different algorithms; in particular, the hybrid applications are highlighted, where novel Artificial Intelligence techniques are coupled with traditional algorithms to solve their shortcomings;
- to provide a performance comparison of different neural approaches used in guidance, navigation and control applications that exist in the literature. In general, it is hard to attribute quantitative metrics to such evaluations, since the applicative scenarios reported in the literature are different. The paper attempts to condense the information into a more qualitative comparison.

The paper is structured as follows: Section 2 presents the foundations of machine learning, deep learning and artificial neural networks, together with a brief theoretical overview of the main training approaches; Section 3 presents an overview of the most used artificial neural networks in the spacecraft dynamics identification, navigation, guidance and control applications. Section 4 reports the applications of several artificial neural networks in the context of spacecraft system identification and guidance, navigation and control systems. Finally, Section 5 draws the conclusions of the paper.

## 2. Machine Learning and Deep Learning

This section provides the theoretical basis of machine learning and deep learning, which are fundamental to understanding the core characteristics of these approaches. The discussion focuses on the domain features that are useful and commonly adopted in specific space-based applications. The research on machine learning (ML) and deep learning (DL) is complex and extremely vast. In order to acquire proper knowledge on the topic, the author suggests referring to [1]. Hereby, only the most relevant concepts are reported in order to contextualize the work developed in the paper. The first important distinction to mark is that between the terms machine learning and deep learning. The highlights of the two approaches are reported in Figure 1.

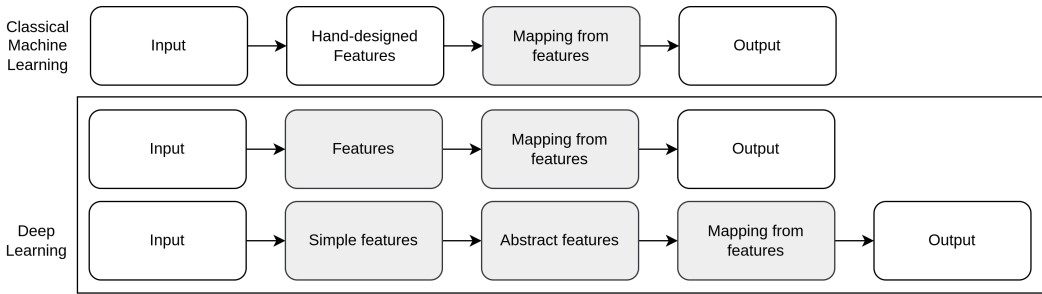

**Figure 1.** Differences between machine learning and deep learning [1].

Machine Learning learns to map input to output given a certain world representation (features) hand-crafted for each task.

Deep learning is a particular kind of machine learning that aims at representing the world as a nested hierarchy of concepts, which are self-detected by the deep learning architecture itself.

The paradigm of ML and DL is to develop algorithms that are data-driven. The information to carry out the task is gathered and derived from either structured or unstructured data. In general, one would have a given experience $\mathcal{E}$, which can be easily thought as a set of data $\mathcal{D} = (x_1, x_2, \ldots, x_n)$. It is possible to divide the algorithms into three different approaches:

- Supervised learning: Given the known outputs $\mathcal{T} = (t_1, t_2, \ldots, t_n)$, we learn to yield the correct output when new datasets are fed.
- Unsupervised Learning: The algorithms exploit regularities in the data to generate an alternative representation used for reasoning, predicting or clustering.
- Reinforcement Learning: Producing actions $\mathcal{A} = (a_1, a_2, \ldots, a_n)$ that affect the environment and receiving rewards $\mathcal{R} = (r_1, r_2, \ldots, r_n)$. Reinforcement learning is all about learning what to do (i.e., mapping situations to actions) so as to maximize a numerical reward.

Even though the boundaries between the approaches are often blurred, the focus of this survey is to discuss algorithms that take advantage and inspiration from supervised and reinforcement learning. For this reason, few additional details are provided for such approaches. Tentative clusters of the different learning approaches and their most used algorithms are reported in Table 2.

**Table 2.** A comprehensive summary for the different learning approaches.

| Learning Approach | Features | Task | Algorithms |
|---|---|---|---|
| Supervised Learning | It learns by exploiting input–output data pairs | Classification | Support Vector Machines, Discriminant Analysis, Nearest Neighbour, Artificial Neural Networks |
| | | Regression | Linear regression, Ensemble methods, Decision Trees, Support Vector Regression, Artificial Neural Networks |
| Unsupervised Learning | It learns to extrapolate patterns and properties of the structure of the dataset | Clustering | K-means, Spectral clustering, Hierarchical clustering, Gaussian Mixture, Hidden Markov Models, Artificial Neural Networks |
| | | Dimensionality Reduction | Principal Component Analysis, Linear Discriminant Analysis, Artificial Neural Networks |
| Reinforcement Learning | It learns the action to undertake based on some inputs, in order to maximize a given reward. | Model-based | Dynamica Programming, Model-given methods, Model-learned methods |
| | | Model-free | Value based methods, Policy-based methods |

*2.1. Supervised Learning*

Supervised learning consists of learning to associate some output with a given input, coherently with the set of examples of inputs $\vec{x}$ and targets $\vec{t}$ [1]. Quite often, the targets $\vec{t}$ are provided by a human supervisor. Nevertheless, supervised learning refers also to approaches in which target states are automatically retrieved by the machine learning model; we use the term this way often throughout this survey. The typical applications of supervised learning are *classification* and *regression*. In a few words, classification is the task of assigning a label to a set of input data from among a finite group of labels. The output is a probability distribution of the likelihood of a certain input of belonging to a certain class. On the other hand, regression aims at modeling the relationships between a certain number of features and a continuous target variable. The regression task is largely employed in supervised learning reported in this survey. Supervised learning is applicable to multiple tasks, both offline [13,14] and online [15,16].

*2.2. Unsupervised Learning*

Unsupervised learning algorithms are fed with a dataset containing many features. The system learns to extrapolate patterns and properties of the structure of this dataset. As reported in [1], in the context of deep learning, the aim is to learn the underlying probability distribution of dataset, whether explicitly as in density estimation or implicitly for tasks such as synthesis or de-noising. Some other unsupervised learning algorithms perform other tasks, such as clustering, which consists of dividing the dataset into separate sets, i.e., clusters of similar experiences and data. The unsupervised learning approach has not yet seen widespread employment in the spacecraft GNC domain.

*2.3. Reinforcement Learning*

Reinforcement learning is learning what to do, how to map observations to actions, so as to maximize a numerical reward signal. The learner is not told which actions to take, but instead must discover which actions yield the most reward by trying them [5].

One of the challenges that arises in reinforcement learning, and not in other kinds of learning, is the trade-off between exploration and exploitation. The agent typically needs to explore the environment in order to learn a proper optimal policy, which determines the required action in a given perceived state. At the same time, the agent needs to exploit such information to actually carry out the task. In the space domain, only for online deployed applications, the balance must be shifted towards exploitation, for practical reasons. Another distinction that ought to be made is between *model-free* and *model-based* reinforcement learning techniques, as shown in Table 3. Model-based methods rely on planning as their primary component, and model-free methods primarily rely on learning. Although there are remarkable differences between these two kinds of methods, there are also great similarities. We call an environmental model whatever information the agent can use to make predictions on what will be the reaction of the environment to a certain action. The environmental model can be known analytically, partially or completely unknown, i.e., to be learned. The model-based algorithms need a representation of the environment. If the agent requires learning the model completely, the exploration is still very important, especially in the first phases of the training. It is worth mentioning that some algorithms start off by mostly exploring, and adaptively trade off exploitation and exploration during optimization, typically ending with very little exploration. For the reasons above, the model-based approach seems to be beneficial in the context of this survey, as it merges the advantages of analytical base models, learning and planning. It is important to report some of the key concepts of reinforcement learning:

- Policy: defines the learning agent's way of behaving at a given time. Mapping from perceived states of the environment to actions to be taken when in those states.
- Reward: at each time step, the environment sends to the reinforcement learning agent a single number called the reward.
- Value Function: the total amount of reward an agent can expect to accumulate in the future, starting from that state.

**Table 3.** Differences between model-based and model-free reinforcement learning. In space, a deterministic representation of a dynamical model is generally available. Nevertheless, some scenarios are unknown (small bodies) or partially known (perturbations).

| Model-Free | Model-Based |
|---|---|
| Unknown system dynamics | Learnt system dynamics |
| The agent is not able to make predictions | The agent makes prediction |
| Need for explorations | More sample efficient |
| Lower computational cost | Higher computational cost |

The standard reinforcement learning theory states that an agent is capable of obtaining a policy which provides the mapping between a set of states $x \in \mathbb{X}$, where $\mathbb{X}$ is the set of possible states, for an action $a \in \mathbb{A}$, where $\mathbb{A}$ is the set of possible actions. The dynamics of the agent are basically represented by a transition probability $p(x_{k+1}|x_k, a_k)$ from one state to another at a given time step. In general, the learned policy can be deterministic $\pi(x_k)$ or stochastic $\pi(a_k|x_k)$, meaning that the control action follows a conditional probability distribution across the states. Every time the agent performs an action, it receives a reward $r(x_k, a_k)$: the ultimate goal of the agent is to maximize the accumulated discounted reward $\mathbb{R} = \sum_{i=k}^{N} \gamma^{i-k} r(x_i, a_i)$ from a given time step $k$ to the end of the horizon $N$, which could be $N = \infty$ in an infinite horizon. The coefficient $\gamma$ is the discount rate, which determines how much more current rewards are to be preferred to future rewards. As mentioned, the value function $V^\pi$ is the total amount of reward an agent can expect to accumulate in the future, in a given state. Note that the value function is obviously associated with a policy:

$$V^\pi(x_k) = \mathbb{E}[\mathbb{R}|x_k, a_k = \pi(x_k)] \tag{1}$$

In most of the reinforcement learning applications, a very important concept is the action-value function $Q^\pi$:

$$Q^\pi(x_k, a_k) = r(x_k, a_k) + \gamma \sum_{x_{k+1}} p(x_{k+1}|x_k, a_k) V^\pi(x_{k+1}) \tag{2}$$

The remarkable difference between it and the value function is the fact that the action-value function tells you the expected cumulative reward at a certain state, given a certain action. The optimal policy is the one that maximizes the value function $\tilde{\pi} = \text{argmax}_\pi V^\pi(x_k)$. In general, an important remark is that reinforcement learning was originally developed for discrete Markov decision processes. This limitation, which is not solved for many RL methods, implies the necessity of discretizing the problem into a system with a finite number of actions. This is sometimes hard to grasp in a domain in which the variables are typically continuous in space and time (think about the states or the control action) or often discretized in time for implementation. Thus, the application of reinforcement learning requires smart ways to treat the problem and dedicated recasting of the problem itself. The reinforcement learning problem has been tackled using several approaches, which can be divided into two main categories: the policy-based methods and the value-based methods. The former ones search for the policy that behaves correctly in a specific environment [17–22]; the latter ones try to value the utility of taking a particular action in a specific state of the environment [23,24]. A common categorization adopted in the literature for identifying the different methods is described below:

- Value-based methods: These methods seek to find optimal value function $V$ and action-value function Q, from which the optimal policy $\pi$ is directly derived. The value-based methods evaluate states and actions. Value-based methods are, for instance, Q-learning, DQN and SARSA [24].
- Policy-based methods: They are methods whose aim is to search for the optimal policy $\pi^*$ directly, which provides a feasible framework for continuous control. The most employed policy-based methods are: advantage actor+critic, cross-entropy methods, deep deterministic policy gradient and proximal policy optimization [17–22].

An additional distinction in reinforcement learning is on-policy and off-policy. On-policy methods attempt to evaluate or improve the policy that is used to make decisions during training, whereas off-policy methods evaluate or improve a policy different from the one used to generate the data, i.e., the experience.

A thorough review, beyond the scope of this paper, is necessary to survey the methods and approaches of reinforcement learning and deep reinforcement learning to space. Some very promising examples were developed in [18–20,23,24], and the most active topics in space applications are reviewed in Section 4.3.

### 2.4. Artificial Neural Networks

Artificial neural networks represent nonlinear extensions to the linear machine learning (or deep learning) models presented in Section 2. A thorough description of artificial neural networks is far beyond the scope of this work. Hereby, the set of concepts necessary to understand the work is reported. In particular, the universal approximation theory is described, which forms the foundation for all the algorithms developed in this paper. The most significant categorization of deep neural networks is into *feedforward* and *recurrent* networks. Deep feedforward networks, also often called multilayer perceptrons (MLPs), are the most common deep learning models. The feedforward network is designed to approximate a given function $f$. According to the task to execute, the input is mapped to an output value. For instance, for a classifier, the network $\mathcal{N}$ maps an input $x$ to a category $y$. A feedforward network defines a mapping $y = \mathcal{N}(x, w)$ and learns the values of the parameters $w$ (weights) that result in the best function approximation. These models are called feedforward because information flows from the input layer, through the intermediate ones, up to the output $y$. Feedback connections are not present in which outputs of the

model are fed back as input to the network itself. When feedforward neural networks are extended to include feedback connections, they are called recurrent neural networks.

The essence of deep learning, and machine learning also, is learning world structures from data. All the algorithms falling into the aforementioned categories are *data-driven*. This means that, despite the possibility of exploiting an analytical representation of the environment, the algorithms need to be fed with structures of data to perform the training. The learning process can be defined as the algorithm by which the free parameters of a neural network are adapted through a process of stimulation by the environment in which it works. The type of learning is the set of instructions for how the parameters are changed, as explained in Section 2. Typically, the following sequence is followed:

1. the environment stimulates the neural network;
2. the neural network makes changes to the free parameters;
3. the neural network responds in a new way according to the new structure.

As one might easily expect, there are several learning algorithms that can consequently be split into different types. It is possible to divide the supervised learning philosophy into batch and incremental learning [4]. Batch learning is suitable for the spatial distribution of data in a stationary environment, meaning that there is no significant time correlation of data, and the environment reproduces itself identically in time. Thus, for such applications, it is possible to gather the data into a whole batch that is presented to the learner simultaneously. Once the training has been successfully completed, the neural networks should be able to capture the underlying statistical behavior of the stationary environment. This kind of statistical memory is used to make predictions exploiting the batch dataset that was presented. This does not mean that batch learning is not capable of transferring knowledge to unseen environments or adapting to real-time applications, as shown in [18,25]. On the other hand, in several applications, the environment is non-stationary, meaning that information signals coming from the environment may vary with time. Batch learning in then inadequate, as there are no means to track and adapt to the varying environmental stimuli. Hence, for on-board learning applications, it is favorable to employ what is called incremental learning (or online or continuous learning) in which the neural network constantly adapts its free parameters to the incoming information in a real-time fashion, as proposed in [15,16,26–30].

### 2.4.1. Universal Approximation Theorem

The universal approximation theorem takes the following classical form [4]. Let $\varphi : \mathbb{R} \to \mathbb{R}$ be a non-constant, bounded and continuous function (called the activation function). Let $I_m$ denote the m-dimensional unit hypercube $[0, 1]^m$. The space of real-valued continuous functions on $I_m$ is denoted by $C(I_m)$. Then, given any $\varepsilon > 0$ and any function $f \in C(I_m)$, there exist an integer $N$, real constants $v_i, b_i \in \mathbb{R}$ and real vectors $w_i \in \mathbb{R}^m$ for $i = 1, \ldots, N$, such that we may define:

$$F(x) = \sum_{i=1}^{N} v_i \varphi \left( w_i^T x + b_i \right) \tag{3}$$

As an approximate realization of the function $f$,

$$|F(x) - f(x)| < \varepsilon \tag{4}$$

For all $x \in I_m$.

### 2.4.2. Training Algorithms

The basis for most of the supervised learning algorithms is represented by *backpropagation*. In general, finding the weights of an artificial neural network means determining the optimal set of variables that minimizes a given loss function. Given $\mathcal{N}$ structured

data, comprising input $x$ and target $t$, one can define the loss function at the output of neuron $j$ for the $p^{th}$ datum presented:

$$\epsilon_j(p) = t_j(p) - y_j(p) \tag{5}$$

where $y_j(p)$ is the output value of the $j^{th}$ output neuron. It is possible to extend this definition to derive a mean indication of the loss function for the complete output layer. We can define a total energy error of the network for the $p^{th}$ presented input–target pair:

$$\mathcal{E} = \frac{1}{2} \sum_{j \in \mathcal{C}_j} \epsilon_j^2(p) \tag{6}$$

where $C_j$ is the set of output neurons of the network. As stated, the total energy error of the network represents the loss function to be minimized during training. Indeed, this function is dependent on all the free parameters of the network, synaptic weights and biases. In order to minimize the energy error function, we need to find those weights that vanish the derivative of the function itself and minimize the argument:

$$(\mathbf{w}, \mathbf{b})^T = argmin\ \mathcal{E}(\vec{w}, \vec{b}) \tag{7}$$

Closed-form solutions are practically never available, thus it is common practice to use iterative algorithms that make use of the derivative of the error function to converge to the optimal value. The back-propagation algorithm is basically a smart way to compute those derivatives, which can then be employed using traditional minimization algorithms, such as [31,32]:

- Batch gradient descent;
- Stochastic gradient descent;
- Conjugate gradient;
- Newton and quasi-Newton methods;
- Levenberg–Marquardt;
- Backpropagation through time.

A slightly different approach, highly tailored to the specific application, is the training through Lyapunov stability-based methods, which will be discussed for the particular application of dynamics reconstruction. Let us consider a simple method that can be applied specifically to sequential learning, in the most common network architecture, but easily extended to batch learning. With reference to Figure 2, the induced local field of neuron $j$, which is the input of the activation function $\phi_j(\cdot)$ at neuron $j$, can be expressed as:

$$v_j(p) = \sum_{i \in \mathcal{C}_i} w_{ji} y_i(p) + b_j \tag{8}$$

where $\mathcal{C}_i$ is the set of neurons that share a connection with layer $j$ and $b_j$ is the bias term of neuron $j$. The output of a neuron is the result of the application of the activation function to the local field $v_j$:

$$y_j(p) = \phi_j(v_j(p)) \tag{9}$$

In gradient-based approaches, the correction to the synaptic weights $w_{ij}$ is performed according to the direction identified by the partial derivatives (i.e., gradient), which can be calculated according to the chain rule as:

$$\frac{\partial \mathcal{E}(p)}{\partial w_{ij}(p)} = \frac{\partial \mathcal{E}(p)}{\partial \epsilon_j(p)} \frac{\partial \epsilon_j(p)}{\partial y_j(p)} \frac{\partial y_j(p)}{\partial v_j(p)} \frac{\partial v_j(p)}{\partial w_{ij}(p)} \tag{10}$$

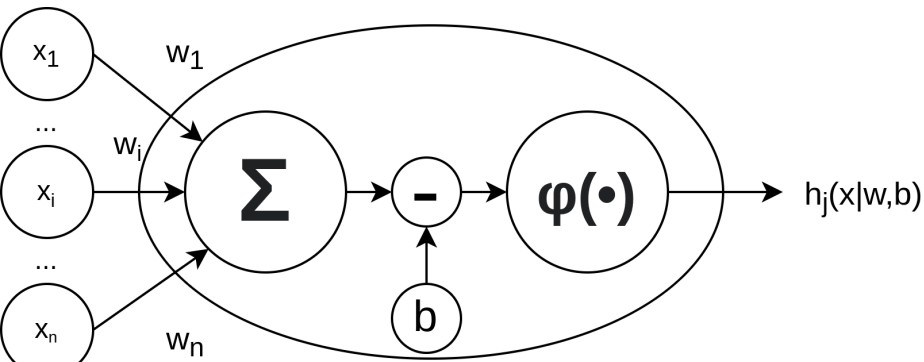

**Figure 2.** Elementary artificial neuron architecture.

Hence, the update to the synaptic weights $\Delta w_{ij}$ is calculated as a *gradient descent* step in the weight space using the derivative of Equation (10):

$$\Delta w_{ij} = -\eta \frac{\partial \mathcal{E}(p)}{\partial w_{ij}(p)} \tag{11}$$

where $\eta$ is the tunable learning-rate parameter. The back-propagation algorithm entails two passages through the network: the *forward pass* and the *backward pass*. The former evaluates the output of the network and the function signal of each neuron. The weights are unaltered during the *forward pass*. The *backward pass* starts from the output layer by passing the loss function back to the input layer, calculating the local gradient for each neuron.

2.4.3. Incremental Learning

Incremental learning stands for the process of updating the weights each time a pair of input–target $((x,t)_p)$ is presented. The two mentioned passes are executed at each step. This is the mode utilized for an online application where the training process can potentially never stop, as the data keep on being presented to the network. In incremental learning, often referred to online learning, the system is trained continuously as new data instances become available. They could be clustered in mini-batches or come as datum by datum. Online learning systems are tuned to set how fast they should adapt to incoming data: typically, such a parameter is referred to as learning rate [33]. A high learning rate means that the system reacts immediately to new data, by adapting itself quickly. However, a high learning rate means that the system will also tend to forget and replace the old data. On the other hand, a low learning rate makes the system more stiff, meaning that it will learn more slowly. Additionally, the system will be less sensitive to noise present in the new data or to mini-batches containing non-representative data points, such as outliers. In addition, Lyapunov-based methods are very suitable for incremental learning due to their inherent step-wise trajectory evaluation of the stability of the learning rule. Two examples of incremental learning system are shown in Figures 3 and 4.

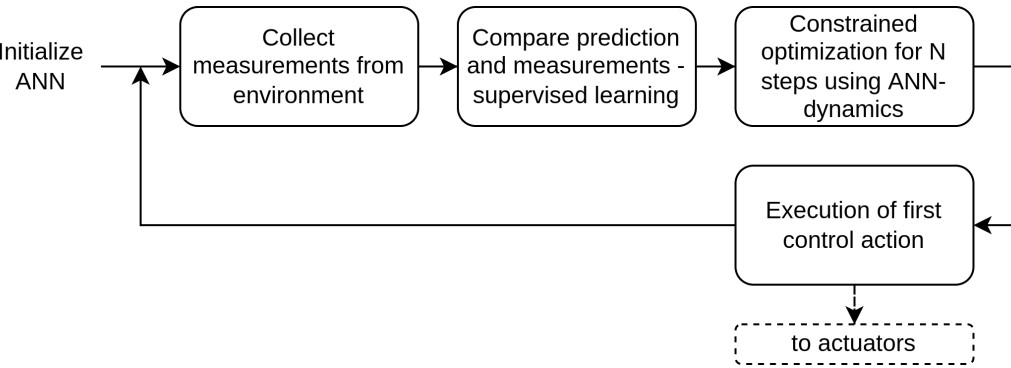

**Figure 3.** Schematics of online incremental learning for spacecraft guidance [27].

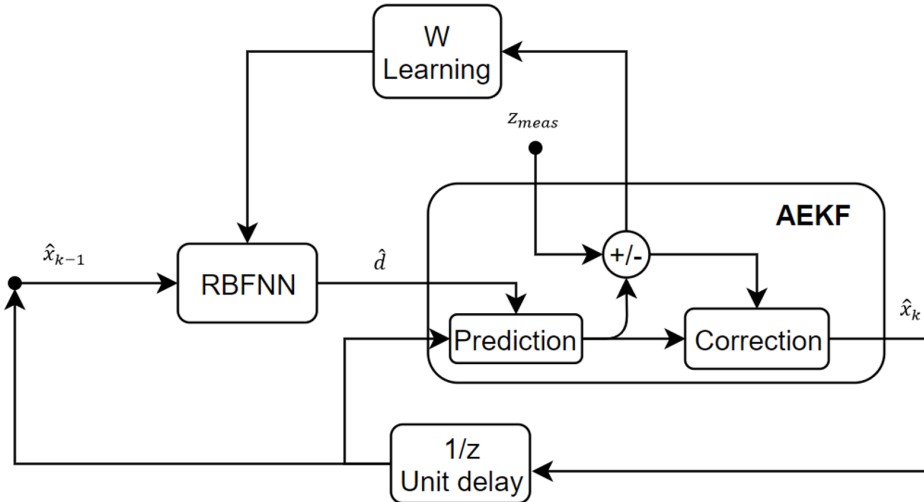

**Figure 4.** Example of incremental learning for spacecraft navigation [29].

### 2.4.4. Batch Learning

Batch learning algorithms execute the weight updates only after all the input–target data are presented to the network [1,33]. One complete presentation of the training dataset is typically called an *epoch*. Hence, after each epoch it is possible to define an average energy error function, which replaces Equation (6) in the back-propagation algorithm:

$$\tilde{\mathcal{E}} = \frac{1}{2\mathcal{N}} \sum_{p=1}^{\mathcal{N}} \sum_{j \in \mathcal{C}_j} \epsilon_j^2(p) \tag{12}$$

The forward and backward passes are performed after each epoch. In batch learning, the system is not capable of learning *while running*. The training dataset consists of all the available data. This generally takes a lot of time and computational effort, given the typical dataset sizes. For this reason, the batch learning is generally performed on the ground. The system that is trained with batch learning first learns offline and then is deployed and runs without updating itself: it just applies what it has learned.

### 2.4.5. Overfitting and Online Sampling

In machine learning, a very common issue encountered in a wrong training process is overfitting. In general terms, overfitting refers to the behavior of a model to perform well on the training data, without generalizing correctly. Complex models, such as deep neural networks, are capable of extracting underlying patterns in the data, but if the dataset is not chosen coherently, the model will most likely form non-existing patterns, or simply patterns that are not useful for generalization [1,33]. The main causes of overfitting can be:

- the training dataset is too noisy;
- the training dataset is too small, which causes sampling noise;
- the training set includes uninformative features.

For instance, in dynamics reconstruction, a high sampling frequency of the state and action is not beneficial for the training. Suppose the first batch of data for learning is very much localized in a given portion of space, say, $\mathcal{R}^9$. Several hyper-surfaces approximate the given transition between $\mathbf{x}_k$ and $\mathbf{x}_{k+1}$. For the sake of explanation, Figure 5 demonstrates the concept in a fictitious 1D model identification. The learning data are enclosed in a restricted region; hence, several curves yield a low loss function in the back-propagation algorithm, but the model is definitely not suitable for generalization. The limitation of the dataset to a bounded and restricted region is not beneficial for identification of dynamics.

Especially in a preliminary learning process, this would drive the neural network to a wrong convergence point.

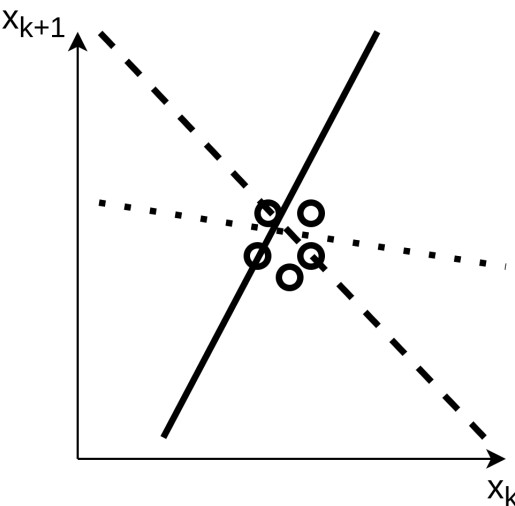

**Figure 5.** 1D dataset for incremental online learning [27].

### 3. Types of Artificial Neural Networks

This section provides insights on the most used architecture in the space domain. The reader is suggested to refer to [1,3] for a comprehensive outlook on the core working principles of neural networks. This survey targets the application of artificial neural networks in space systems; thus, only relevant architectures, namely, those actually investigated and implemented in spacecraft GNC systems, are detailed. A summary of the most common neural network architectures is reported in Table 4.

**Table 4.** Most popular activation functions used in MLP.

| Activation Function | $\Phi$ | $\Phi'$ | Codomain | Ref. |
|---|---|---|---|---|
| Hyperbolic Tangent | $\tanh x$ | $1 - \tanh x$ | $(-1, 1)$ | [27,34] |
| Sigmoid | $1/(1 + e^{-x})$ | $\Phi(x)(1 - \Phi(x))$ | $(0, 1)$ | [34,35] |
| ReLu | $\max(0, x)$ | $\max(0, x)$ | $[0, \infty)$ | [27,36,37] |
| Signum | $\text{sgn}(x)$ | $2\delta_0$ | $[-1, 1]$ | |
| Heaviside step | $(\text{sgn}(x) + 1)/2$ | $\delta_0$ | $[0, 1]$ | [38] |
| Softmax | $e^{x_i} / \sum_j^n e^{x_j}$ | $\Phi_i(\delta_{ij} - \Phi_j)$ | $(0, 1)$ | output |

#### 3.1. Feed-Forward Networks

Feedforward neural networks (FFNN) are the oldest and most common network architecture, and they form the fundamental basis for most of the deep learning models. The term feedforward refers to the information flow that the network possesses: the network is evaluated starting from $\vec{x}$ to the output $\vec{y}$. The network generates an acyclic graph. Two important design parameters to take into account when designing a neural network are:

- **Depth**: Typical neural networks are actually nested evaluations of different functions, commonly named input, hidden and output *layers*. In practical applications, low-level features of the dataset are captured by the initial layers up to high-level features learned in the subsequent layer, all the way to the output layer.
- **Width**: Each layer is generally a vector valued function. The size of this vector valued function, represented by the number of neurons, is the width of the model or layer.

Feedforward networks are a conceptual stepping stone on the path to recurrent networks [1,3,4].

### 3.1.1. Multilayer Perceptron

The multilayer perceptron is the most used deep model that is developed to build an approximation of a given function $\tilde{f}$ [1,3,4]. The network $\vec{y} = \mathcal{N}(\vec{x}, \vec{w})$ defines the mapping between input and output and learns the optimal values of the weights $\vec{w}$ that yield the best function approximation. The elementary unit of the MLP is the neuron. With reference to Figure 2, the induced local field of neuron $j$, which is the input of the activation function $\phi_j(\cdot)$ at neuron $j$, can be expressed as:

$$v_j(p) = \sum_{i \in \mathcal{C}_i} w_{ji} y_i(p) + b_j \tag{13}$$

where $\mathcal{C}_i$ is the set of neurons that share a connection with layer $j$; $b_j$ is the bias term of neuron $j$. The output of a neuron is the result of the application of the activation function to the local field $v_j$:

$$y_j(p) = \phi_j(v_j(p)) \tag{14}$$

The activation function (also known as unit function or transfer function) performs a non-linear transformation of the input state. The most common activation functions are reported in Table 5. Among the most commonly used, at least in spacecraft related applications, are the hyperbolic tangent and the ReLu unit. The softmax function is basically an indirect normalization: it maps a $n$-dimensional vector $x$ into a normalized n-dimensional output vector. Hence, the output vector values represent probabilities for each of the input elements. The softmax function is often used in the final output layer of a network; therefore, it is generally different from the activation functions used in each hidden layer. For the sake of completeness, a perceptron is originally defined as a neuron that has the Heaviside function as the activation function. An example of an MLP is reported in Section 4. The MLP has been successfully applied in classification, regression and function approximations.

### 3.1.2. Radial-Basis Function Neural Network

A radial-basis-function neural network is a single-layer shallow network whose neurons are Gaussian functions. This network architecture possesses a quick learning process, which makes it suitable for online dynamics identification and reconstruction. The highlights of the mathematical expression of the RBFNN are reported here for clarity. For a generic state input $\vec{\delta\chi} \in \mathbb{R}^n$, the components of the output vector $\vec{\gamma} \in \mathbb{R}^j$ of the network are:

$$\gamma_l(\vec{\delta\chi}) = \sum_{i=1}^{m} w_{il} \Phi_i(\vec{\delta\chi}) \tag{15}$$

In a compact form, the output of the network can be expressed as:

$$\gamma(\vec{\delta\chi}) = W^T \Phi(\vec{\delta\chi}) \tag{16}$$

where $\vec{W} = [w_{il}]$ for $i = 1, ..., m$, $l = 1, ..., j$ is the trained weight matrix and $\Phi(\vec{\delta\chi}) = [\Phi_1(\delta\chi) \; \Phi_2(\delta\chi) \; \cdots \; \Phi_m(\delta\chi)]^T$ is the vector containing the output of the radial basis functions, evaluated at the current system state. The RBF network learns to designate the input to a center, and the output layer combines the outputs of the radial basis function and weight parameters to perform classification or inference. Radial basis functions are suitable for classification, function approximation and time series prediction problems. Typically, the RBF network has a simpler structure and a much faster training process with respect to MLP, due to the inherent capability of approximating nonlinear functions using shallow architecture. As one could note, the main difference in the RBFNN with respect to the MLP is that the kernel is a nonlinear function of the information flow: in other words, the actual input to the layer is the nonlinear radial function $\Phi(\delta\chi)$ evaluated at the input

data $\delta\chi$, most commonly Gaussian ones. The most used radial-basis functions that can be used and that are found in space applications are [15,29,39]:

$$\text{Gaussian}: \Phi(r) = e^{-\frac{(r-c)^2}{2\sigma^2}}$$

$$\Phi(r) = \frac{1}{(\sigma^2 + r^2)^\alpha}$$

$$\text{Linear}: \Phi(r) = r$$

$$\text{Thin-plate Spline}: \Phi(r) = r^2 ln(r)$$

$$\text{Logistic Function}: \Phi(r) = \frac{1}{1 + e^{(r/\sigma^2)-\theta}}$$

where $r$ is the distance from the origin, $c$ is the center of the RBF, $\sigma$ is a control parameter to tune the smoothness of the basis function and $\theta$ is a generic bias. The number of neurons is application-dependent, and it shall be selected by trading off the training time and approximation [29], especially for incremental learning applications. The same consideration holds for the parameters $\eta = \frac{1}{\sigma}$, which impact the shape of the Gaussian functions. A high value for $\eta$ sharpens the Gaussian bell-shape, whereas a low value spreads it on the real space. On the one hand, a narrow Gaussian function increases the responsiveness of the RBF network; on the other hand, in the case of limited overlapping of the neuronal functions due to overly narrow Gaussian bells, the output of the network vanishes. Hence, ideally, the parameter $\eta$ is selected based on the order of magnitude of the exponential argument in the Gaussian function. The output of the neural network hidden layer, namely, the radial functions evaluation, is normalized:

$$\vec{\Phi}_{norm}(\delta\chi) = \frac{\Phi(\delta\chi)}{\sum_{i=1}^{m} \Phi_i(\delta\chi)} \tag{17}$$

The classic RBF network presents an inherent localized characteristic, whereas the normalized RBF network exhibits good generalization properties, which decreases the curse of dimensionality that occurs with classic RBFNN [39]. A schematic of a RBFNN is reported in Figure 6.

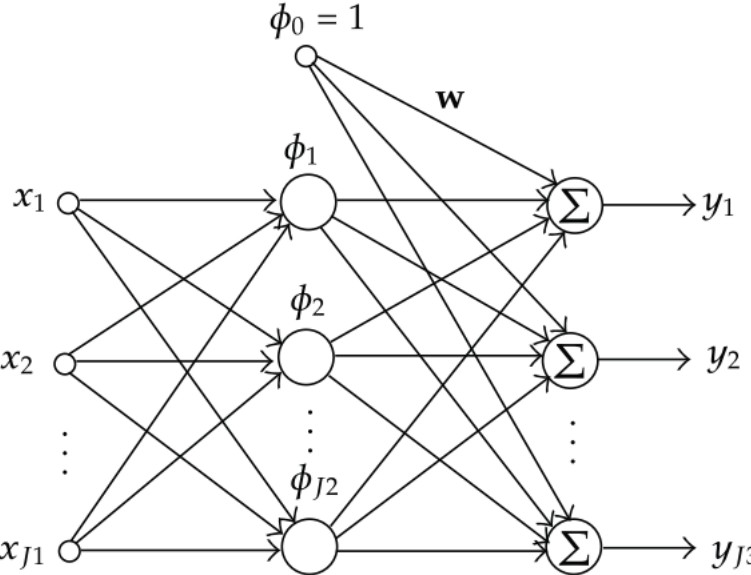

**Figure 6.** Architecture of the RBF network. The input, hidden and output layers have J1, J2 and J3 neurons, respectively [39].

**Table 5.** Summary of most common ANN architecture used in space dynamics, guidance, navigation and control domain. The training types are supervised (S), unsupervised (U) and reinforcement learning (R).

| Network Type | Architecture | Training | Algorithm | Space Applications |
|---|---|---|---|---|
| Feedforward | MLP | S/R | Backpropagation | Dynamics approximation, value function approximation |
| | RBFNN | S/U/R | Backpropagation/ Lyapunov/K-means clustering | Dynamics approximation, regression, time-series prediction |
| | AE | U | Backpropagation | Dimensionality reduction, state-space modelling, data encoding, anomaly detection |
| | CNN | S | Backpropagation | Feature detection, image classification, vision-based navigation |
| Recurrent | LRNN | S/R | Backpropagation through time | Dynamics approximation, time-series prediction |
| | NARX | S/R | Backpropagation through time | Dynamics approximation, time-series prediction |
| | HNN | S | Backpropagation through time | Combinatorial optimization, system identification |
| | LSTM | S/R | Backpropagation through time | Time-series prediction, dynamics approximation |
| | GRU | S/R | Backpropagation through time | Time-series prediction, dynamics approximation, anomaly detection |

### 3.1.3. Autoencoders

The autoencoder is a particular feedforward neural network trained using unsupervised learning. The autoencoder learns to reproduce the unit mapping from a certain information input vector $\vec{I} \in \mathcal{R}^{n \times n}$ to $\vec{I}$ itself. The topological constraint dictates that the number of neurons in the next layer must be lower than the previous one. Such a constraint forces the network to learn a description of the input vector that belongs to the lower-dimensional space of the subsequent layers without losing information. The amount of information lost while encoding a downsizing the input vector is measured by the fitting discrepancy between the input and the reconstructed vector $\vec{I}$ [31,32,40]. The desired lower-dimensional vector concentrating the information contained in the input vector is the layer at which the network starts growing again; see Figure 7. It is important to note that the structure of an autoencoder is exactly the same as the MLP, with the additional constraint of having the same numbers of input and output nodes.

The autoencoders are widely used for unsupervised applications: typically, they are used for denoising, dimensionality reduction and data representation learning.

### 3.1.4. Convolutional Neural Networks

Feedforward networks are of extreme importance to machine learning applications in the space domain. A specialized kind of feedforward network, often referred as a stand-alone type, is the convolutional neural network (CNN) [11]. Convolutional networks are specifically tailored for image processing; for instance, CNNs are used for object recognition, image segmentation and classification. The main reason why traditional feedforward networks are not suitable for handling images is due to the fact that one

image can be thought of as a large matrix array. The number of weights, or parameters, to efficiently process large two-dimensional images (or three if more image channels are involved) quickly explodes as the image resolution grows. In general, given a network of width W and depth D, the number of parameters $n_w$ for a fully connected network is $n_w \sim DW^2 + W$. For instance, a low resolution image $\vec{I} \in \mathcal{R}^{32 \times 32}$ has a width of $W^2$, by simply unrolling the image into a 1D array: this means that $n_w \, 10^6$. A high resolution image, e.g., $\vec{I} \in \mathcal{R}^{1024 \times 1024}$, quickly reaches $n_w \sim 10^{12}$. This shortcoming results in complex training procedures, very much subject to overfitting. The convolutional neural network paradigm stands for the idea of reducing the number of parameters starting from the main assumptions:

1. Low-level features are local;
2. Features are translationally invariant;
3. High-level features are composed of low-level features.

Such assumptions allow a reduction in the number of parameters while achieving better generalization and improved scalability to large datasets. Indeed, instead of using fully connected layers, a CNN uses local connectivity between neurons; i.e., a neuron is only connected to nearby neurons in the next layer [32]. The basic components of a convolutional neural network are:

- Convolutional layer: the convolutional layer is core of the CNN architecture. The convolutional layer is built up by neurons which are not connected to every single neuron from the previous layer but only to those falling inside their receptive field. Such architecture allows the network to identify low-level features in the very first hidden layer, whereas high-level features are combined and identified at later stages in the network. A neuron's weight can be thought of as a small image, called the *filter* or *convolutional kernel*, which is the size of the receptive field. The convolutional layer mimics the convolution operation of a convolutional kernel on the input layer to produce an output layer, often called the *feature map*. Typically, the neurons that belong to a given convolutional layer all share the same convolutional kernel: this is referred to as *parameter sharing* in the literature. For this reason, the element-wise multiplication of each neuron's weight by its receptive field is equivalent to a pure convolution in which the kernel slides across the input layer to generate the feature map. In mathematical terms, a convolutional layer, with convolutional kernel $\vec{W}$, operating on the previous layer $\vec{I}$ (being either an intermediate feature map or the input image), performs the following operation:

$$f_{i,j} = (\vec{I} * \vec{W}) \tag{18}$$

where $f_{i,j}$ is the $(i, j)$ position of the output feature map.
- Activation layer: An activation function is utilized as a decision gate that aids the learning process of intricate patterns. The selection of an appropriate activation function can accelerate the learning process [11]. The most common activation functions are the same as those used for the MLP and are presented in Table 4.
- Pooling layer: The objective of a pooling layer is to sub-sample the input image or the previous layer in order to reduce the computational load, the memory usage and the number of parameters, which prevents overfitting while training [11,33]. The pooling layer works exactly with the same principle of the receptive field. However, a pooling neuron has no weights; hence, it aggregates the inputs by calculating the maximum or the average within the receptive field as output.
- Fully-connected layer: Similarly to MLP as for traditional CNN architectures, a fully connected layer is often added right before the output layer to further capture non-linear relationships of the input features [11,32]. The same considerations discussed for MLP hold for CNN fully connected layers.

An example of a CNN architecture is shown in Figure 8.

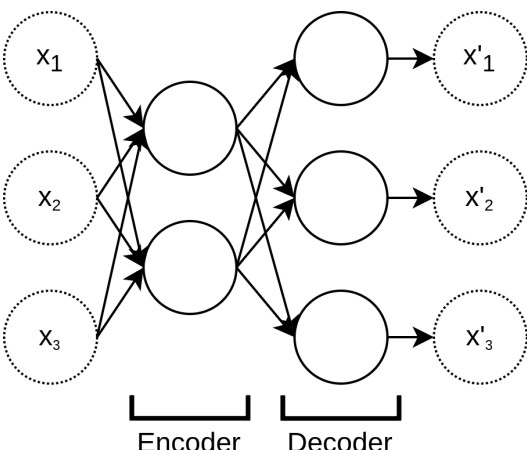

**Figure 7.** Basic autoencoder structure.

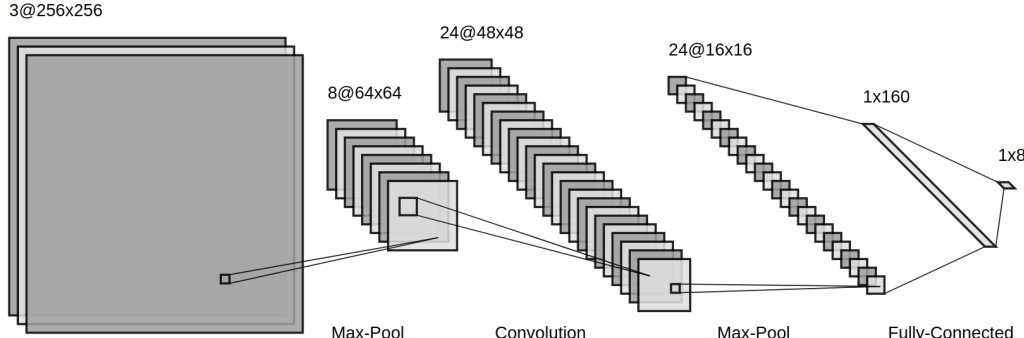

**Figure 8.** Example CNN architecture with convolutional, max-pooling and fully connected layers.

*3.2. Recurrent Neural Networks*

Recurrent neural networks comprise all the architectures that present at least one feedback loop in their layer interactions [1]. A subdivision that is seldom used is between finite and infinite impulse recurrent networks. The former is given by a directed acyclic graph (DAG) that can be unrolled in time and replaced with a feedforward neural network. The latter is a directed cyclic graph (DCG) that cannot be unrolled and replaced similarly [32]. Recurrent neural networks have the capability of handling time-series data efficiently. The connections between neurons form a directed graph, which allows internal state memory. This enables the network to exhibit temporal dynamic behaviors.

3.2.1. Layer-Recurrent Neural Network

The core of the layer-recurrent neural network (LRNN) is similar to that of the standard MLP [1]. This means that the same considerations for model depth, width and activation functions hold in the same manner. The only addition is that in the LRNN, there is a feedback loop with a single delay around each layer of the network, except for the last layer. A schematic of the LRNN is sketched in Figure 9.

3.2.2. Nonlinear Autoregressive Exogenous Model

The nonlinear autoregressive exogenous model is an extension of the LRNN that uses the feedback coming from the output layer [41]. The LRNN owns dynamics only at the input layer. The nonlinear autoregressive network with exogenous inputs (NARX) is a recurrent dynamic network with feedback connections enclosing several layers of the network. The NARX model is based on the linear ARX model, which is commonly used in time-series modeling. The defining equation for the NARX model is

$$y_k = \mathcal{N}(y_{k-1}, y_{k-2}, \ldots, y_{k-n}, u_{k-1}, u_{k-2}, \ldots, u_{k-n}) \tag{19}$$

where $y$ is the network output and $u$ is the exogenous input, as shown in Figure 10. Basically, it means that the next value of the dependent output signal $y$ is regressed on previous values of the output signal and previous values of an independent (exogenous) input signal. It is important to remark that, for a one tap-delay NARX, the defining equation takes the form of an autonomous dynamical system.

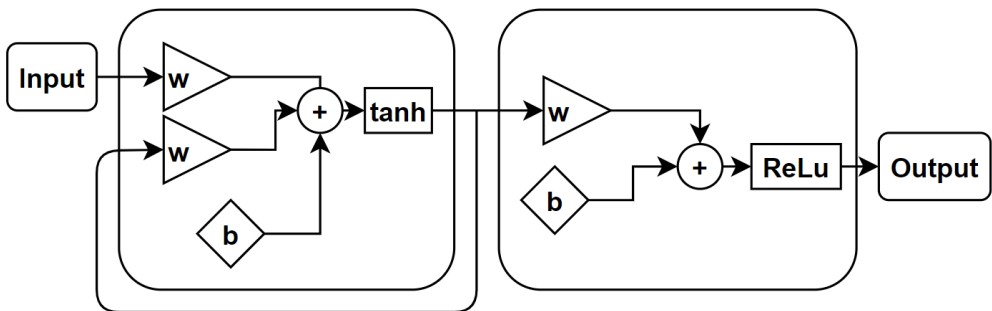

**Figure 9.** Schematic of a layer-recurrent neural network. The feedback loop is a tap-delayed signal rout.

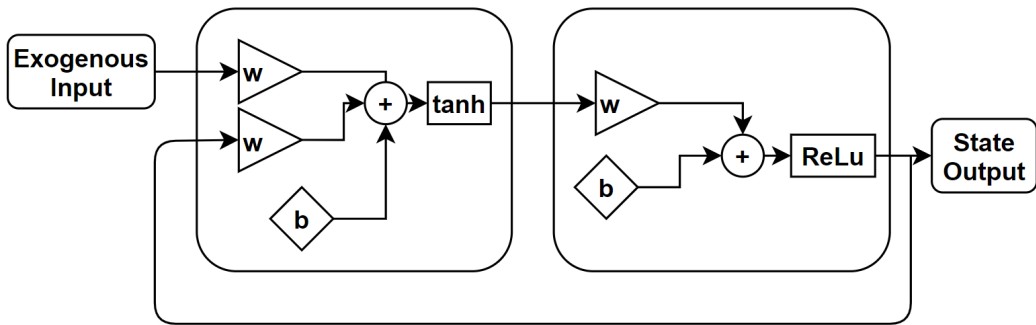

**Figure 10.** Schematic of a nonlinear autoregressive exogenous model. The feedback loop is a tap-delayed signal rout [27].

### 3.2.3. Hopfield Neural Network

The formulation of the network was due to Hopfield [42], but the formulation by Abe [43] is reportedly the most suited for combinatorial optimization problems [44], which are of great interest in the space domain. For this reason, here the most recent architecture is reported. A schematic of the network architecture is shown in Figure 11.

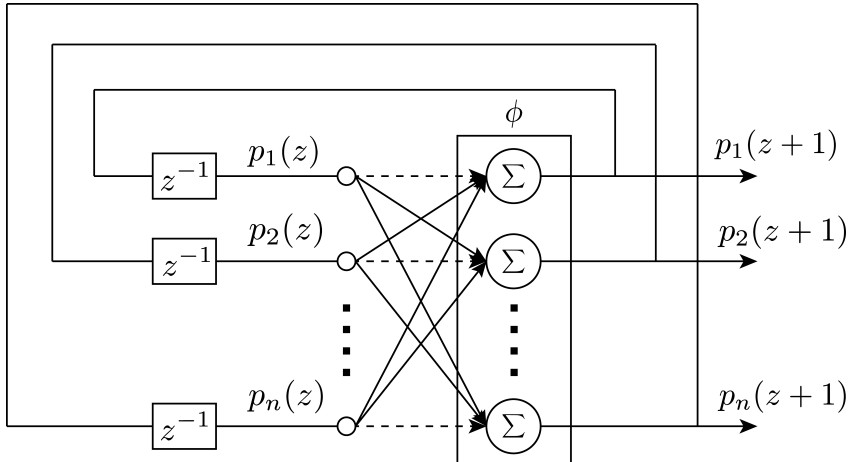

**Figure 11.** The Hopfield neural network structure [44].

In synthesis, the dynamics of the *i*-th out of $N$ neurons is written as:

$$\frac{dp_i}{dt} = \sum_{j=1}^{N} w_{ij}s_j - b_i \tag{20}$$

where $p_i$ is the total input of the i-th neuron; $w_{ij}$ and $b_i$ are parameters corresponding, respectively, to the synaptic efficiency associated with the connection from neuron $j$ to neuron $i$ and the bias of the neuron $i$. The term $s_i$ is basically the equivalent of the activation function:

$$s_i = c_i \tanh \frac{p_i}{\beta} \tag{21}$$

where $\beta > 0$ is a user-defined coefficient, and $c_i$ is the user-defined amplitude of the activation function. The recurrent structure of the network entails the dynamics of the neurons; hence, it would be more correct to refer to $p(t)$ and $s(t)$ as functions of time or any other independent variable. An important property of the network, which will be further discussed in the application for parameter identification, is that the Lyapunov stability theorem can be used to guarantee its stability. Indeed, since a Lyapunov function exists, the only possible long-term behavior of the neurons is to asymptotically approach a point that belongs to the set of fixed points, meaning where $\frac{d\mathcal{V}}{dt} = 0$, $\mathcal{V}$ being the Lyapunov function of the system, in the form:

$$\mathcal{V} = -\frac{1}{2}\sum_{i=1}^{N}\sum_{j=1}^{N} w_{ij}s_is_j + \sum_{i=1}^{N} b_is_i = -\frac{1}{2}\vec{s}^T \mathbb{W}\vec{s} + \vec{s}^T\vec{b} \tag{22}$$

where the right-hand term is expressed in a compact form, with $\vec{s}$, the vector of $s$ neuron states, and $\vec{b}$, the bias vector. A remarkable property of the network is that the trajectories always remain within the hypercube $[-c_i, c_i]$ as long as the initial values belong to the hypercube too [44,45]. For implementation purposes, the discrete version of the HNN is employed, as was done in [44,46].

### 3.2.4. Long Short-Term Memory

The long-short term memory network is a type of recurrent neural network widely used for making predictions based on times series data. LSTM, first proposed by Hochreiter [47], is a powerful extension of the standard RNN architecture because it solves the issue of *vanishing gradients*, which often occur in network training. In general, the repeating module in a standard RNN contains a single layer. This means that if the RNN is unrolled, you can replicate the recurrent architecture by juxtaposing a single layer of nuclei. LSTMs can also be unrolled, but the repeating module owns four interacting layers or *gates*. The basic LSTM architecture is shown in Figure 12.

The core idea is that the cell state lets the information flow: it is modified by the three gates, composed of a sigmoid neural net layer and a point-wise multiplication operation. The sigmoid layer of each gate outputs a value $\in [0, 1]$ that defines how much of the core information is let through. The basic components of the LSTM network are summarized here:

- Cell state (C): The cell state is the core element. It conveys information through different time steps. It is modified by linear interactions with the gates.
- Forget gate ($f$): The forget gate is used to decide which information to let through. It looks at the input $x_k$ and output of the previous step $y_{k-1}$ and yields a number $\in [0, 1]$ for each element of the cell state. In compact form:

$$f = \sigma(\mathbb{W}_f \cdot [y_{k-1}, x_k] + \vec{b}_f] \tag{23}$$

- Input gate ($i$): The input gate is used to decide what piece of information to include in the cell state. The sigmoid layer is used to decide on which value to update, whereas

the *tanh* describes the entities for modification, namely, the values. It then generates a new estimate for the cell state $\tilde{C}$:

$$
\begin{aligned}
i &= \sigma(\mathbb{W}_i \cdot [y_{k-1}, x_k] + \vec{b}_i) \\
\tilde{C}_k &= \tanh(\mathbb{W}_c \cdot [y_{k-1}, x_k] + \vec{b}_c)
\end{aligned}
\tag{24}
$$

- Memory gate: The memory gate multiplies the old cell state with the output of the forget gate and adds it to the output of the input gate. Often, the memory gate is not reported as a stand-alone gate, due to the fact that it represents a modification of the cell state itself, without a proper sigmoid layer:

$$
C_k = f \odot C_{k-1} + i \odot \tilde{C}_k
\tag{25}
$$

- Output gate: The output gate is the final step that delivers the actual output of the network $y_k$, a filtered version of the cell state. The layer operations read:

$$
\begin{aligned}
o &= \sigma(\mathbb{W}_o \cdot [y_{k-1}, x_k] + \vec{b}_o) \\
y_k &= o \odot \tanh C_k
\end{aligned}
\tag{26}
$$

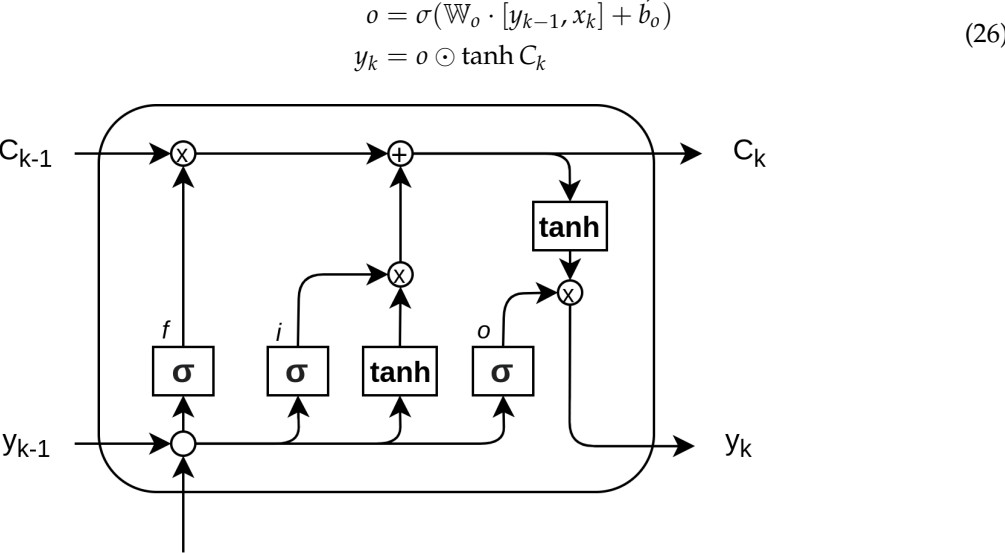

**Figure 12.** The core components of the LSTM are the *cell* (*C*), the *input* gate (*i*), the *output* gate (*o*) and the *forget* gate (*f*) [47].

In contrast to deep feedforward neural networks, having a recurrent architecture, LSTMs contain feedback connections. Moreover, LSTMs are well suited not only for processing single data points, such as input vectors, but efficiently and effectively handle sequences of data. For this reason, LSTMs are particularly useful for analyzing temporal series and recurrent patterns.

### 3.2.5. Gated Recurrent Unit

The gated recurrent unit (GRU) was proposed by Cho [48] to make each recurrent unit adaptively capture dependencies of different time scales. Similarly to the LSTM unit, the GRU has gating units that modulate the flow of information inside the unit, but without having a separate memory cells [48,49]. The basic components of GRU share similarities with LSTM. Traditionally, different names are used to identify the gates:

- Update gate (*u*): The update gate defines how much the unit updates its value or content. It is a simple layer that performs:

$$
u = \sigma(\mathbb{W}_u \cdot [y_{k-1}, x_k] + \vec{b}_u)
\tag{27}
$$

- Reset gate *r*: The reset gate effectively makes the unit process the input sequence, allowing it to forget the previously computed state:

$$r = \sigma(\mathbb{W}_r \cdot [y_{k-1}, x_k] + \vec{b}_r) \tag{28}$$

The output of the network is calculated through a two-step update, entailing a candidate update activation $\tilde{y}_k$ calculated in the activation *h* layer and the output $y_k$:

$$\begin{aligned} \tilde{y}_k &= \tanh \mathbb{W}_h \cdot [y_{k-1}, x_k] + \vec{b}_h \\ y_k &= (1 - u) \odot y_{k-1} + u \odot \tilde{y}_k \end{aligned} \tag{29}$$

A schematic of GRU network is reported in Figure 13.

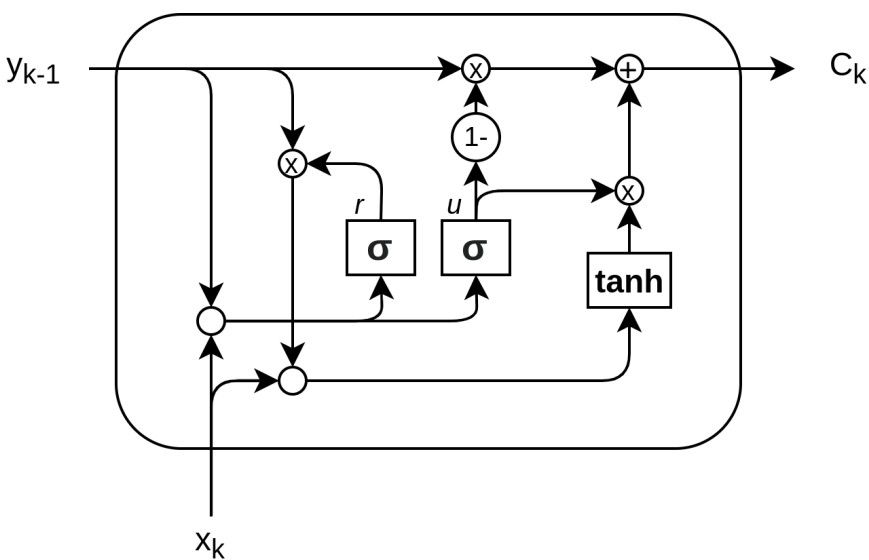

**Figure 13.** The core components of the GRU are the *reset* gate (*r*) and the *update* gate (*u*) coupled with the activation output composed of the tanh layer.

### 3.3. Spiking Neural Networks

Spiking neural networks (SNN) are becoming increasingly interesting to the space domain due to their low-power and energy efficiency. Indeed, small satellite missions entail low-computational-power devices and in general lower system power budgets. For this reason, SNNs represent a promising candidate for the implementation of neural-based algorithms used for many machine learning applications: among those, the scene classification task is of primary importance for the space community. SNNs are the third generation of artificial neural networks (ANNs), where each neuron in the network uses discrete spikes to communicate in an event-based manner. SNNs have the potential advantage of achieving better energy efficiency than their ANN counterparts. While generally a loss of accuracy in SNN models is reported, new algorithms and training techniques can help with closing the gap in accuracy performance while keeping the low-energy profile. Spiking neural networks (SNNs) are inspired by information processing in biology. The main difference is that neurons in ANNs are mostly non-linear but continuous function evaluations that operate synchronously. On the other hand, biological neurons employ asynchronous spikes that signal the occurrence of some characteristic events by digital and temporally precise action potentials. In recent years, researchers from the domains of machine learning, computational neuroscience, neuromorphic engineering and embedded systems design have tried to bridge the gap between the big success of DNNs in AI applications and the promise of spiking neural networks (SNNs) [50–52]. The large spike sparsity and simple synaptic operations (SOPs) in the network enable SNNs to outperform ANNs in terms of energy efficiency. Nevertheless, the accuracy performance,

especially in complex classification tasks, is still superior for deep ANNs. In the space domain, the SNNs are at the earliest stage of research: mission designers strive to create algorithms characterized by great computational efficiency and low power applications; hence, the SNNs represent an interesting opportunity that ought to be mentioned in this review, although they are not yet applied to guidance, navigation and control applications. Finally, SNNs on neuromorphic hardware exhibit favorable properties such as low power consumption, fast inference and event-driven information processing. This makes them interesting candidates for the efficient implementation of deep neural networks particularly utilized in image classification.

The most peculiar feature of SNNs is that the neurons possess temporal dynamics: typically, an electrical analogy is used to describe their behavior. Each neuron has a voltage potential that builds up depending on the input current that it receives. The input current is generally triggered by the spikes the neuron receives. A schematic of the neuron parameters can be seen in Figures 14 and 15. There are numerous neural architectures that combine these notions into a set of mathematical equations; nevertheless, the two most common alternatives are the integrate-and-fire neuron and the leaky-integrate-and-fire neuron.

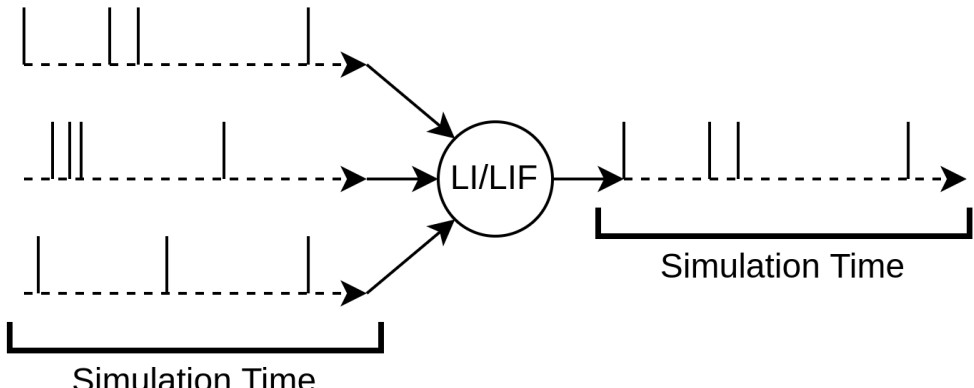

**Figure 14.** Architecture of a simple spiking neuron. Spikes are received as inputs, which are then either integrated or summed depending on the neuron model. The output spikes are generated when the internal state of the neuron reaches a given threshold.

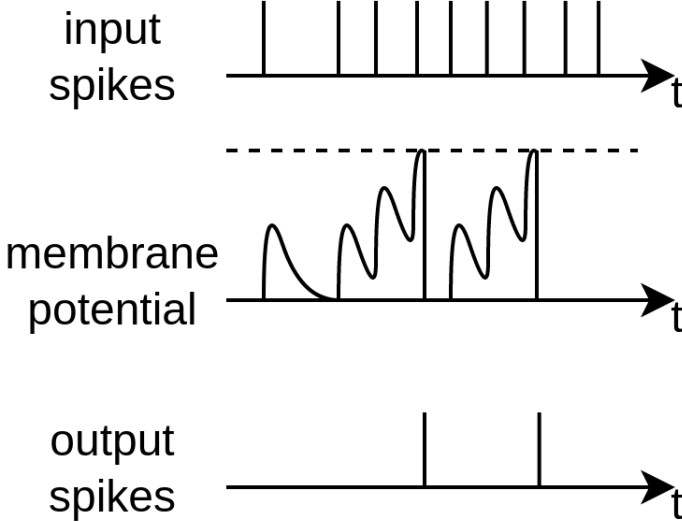

**Figure 15.** An illustrative schematic depicting the membrane potential reaching the threshold, which generates output spikes [51].

3.3.1. Types of Neurons

- Integrate and fire (IF): The IF neuron model assumes that spike initiation is governed by a voltage threshold. When the synaptic membrane reaches and exceeds a certain threshold, the neuron fires a spike and the membrane is set back to the resting voltage $V_{rest}$. In mathematical terms, its simplest form reads:

$$C\frac{dV(t)}{dt} = i(t) \tag{30}$$

- Leaky integrate and fire (LIF): The LIF neuron is a slightly modified version of the IF neuron model. Indeed, it entails an exponential decrease in membrane potential when not excited. The membrane charges and discharges exponentially in response to injected current. The differential equation governing such behavior can be written as:

$$C\frac{dV(t)}{dt} + \lambda V(t) = i(t) \tag{31}$$

where $\lambda$ is the leak conductance and $V$ is again the membrane potential with respect to the rest value.

As mentioned, the list is not extensive, and the reader is suggested to refer to [53] for a comprehensive review of neuron models.

3.3.2. Coding Schemes

The transition between dense data and sparse spiking patterns requires a coding mechanism for input coding and output decoding. For what concerns the input coding, the data can be transformed from dense to sparse spikes in different ways, among which the most used are:

- Rate coding: it converts the input intensity into a firing rate or spike count;
- Temporal (or latency) coding: it converts the input intensity to a spike time or relative spike time.

Similarly, in output decoding, the data can be transformed from sparse spikes to network output (such as classification class) in different ways, among which the most used are:

- Rate coding: it selects the output neuron with the highest firing rate, or spike count, as the predicted class;
- Temporal (or latency) coding: it selects the output neuron that fires first, or before a given threshold time, as the predicted class

Roughly speaking, the current literature agrees on specific advantages for both the coding techniques. On one hand, the rate coding is more error tolerant given the reduced sparsity of the neuron activation. Moreover, the accuracy and learning convergence have shown superior results in rate-based applications so far. On the other hand, given the inherent sparsity of the encoding-decoding scheme, latency-based approaches tend to outperform the rate-based architectures in inference, training speed and, above all, power consumption.

## 4. Applications in Space

This section provides an overview of the space domain tasks that are currently being investigated by the research community. The paper highlights the characteristics that are peculiar to GNC algorithms, referencing other domains' literature when needed. In addition, a short paragraph on the challenges of dataset availability and data validation is presented.

### 4.1. Identification of Neural Spacecraft Dynamics

The capability of using an ANN to approximate the underlying dynamics of a spacecraft is used to enhance the on-board model accuracy and flexibility to provide the spacecraft with a higher degree of autonomy. There are different approaches in the literature that could be adopted to tackle the system identification and dynamics reconstruction task, as show in Figure 16. In the following section, the three analyzed methods are described; recall the universal approximation theorem reported in Section 2.

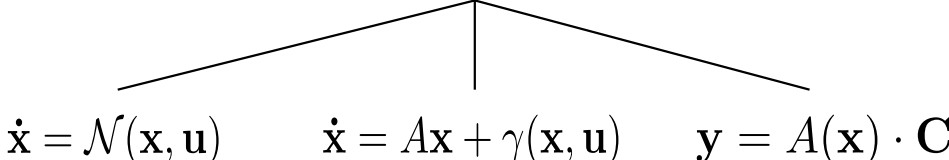

# Dynamics Reconstruction

$$\dot{\mathbf{x}} = \mathcal{N}(\mathbf{x}, \mathbf{u}) \qquad \dot{\mathbf{x}} = A\mathbf{x} + \gamma(\mathbf{x}, \mathbf{u}) \qquad \mathbf{y} = A(\mathbf{x}) \cdot \mathbf{C}$$

**Figure 16.** Identification of system dynamics: different approaches to reconstructing system dynamical behavior.

#### 4.1.1. Fully Neural Dynamics Learning

The dynamical model of a system delivers the derivative of the system state, given the actual system state and external input. Such an input–output structure can be fully approximated by an artificial neural network model. The dynamics are entirely encapsulated in the weights and biases of the network $\mathcal{N}$. The neural network is stimulated by the actual state and the external output. In turn, the time derivative of the state, or simply the system state at the next discretization step, is yielded as output, as shown in Figure 17:

$$\vec{\dot{x}} = \mathcal{N}(\vec{x}, \vec{u}) \rightarrow \vec{x}_{k+1} = \tilde{\mathcal{N}}(\vec{x}_k, \vec{u}_k) \tag{32}$$

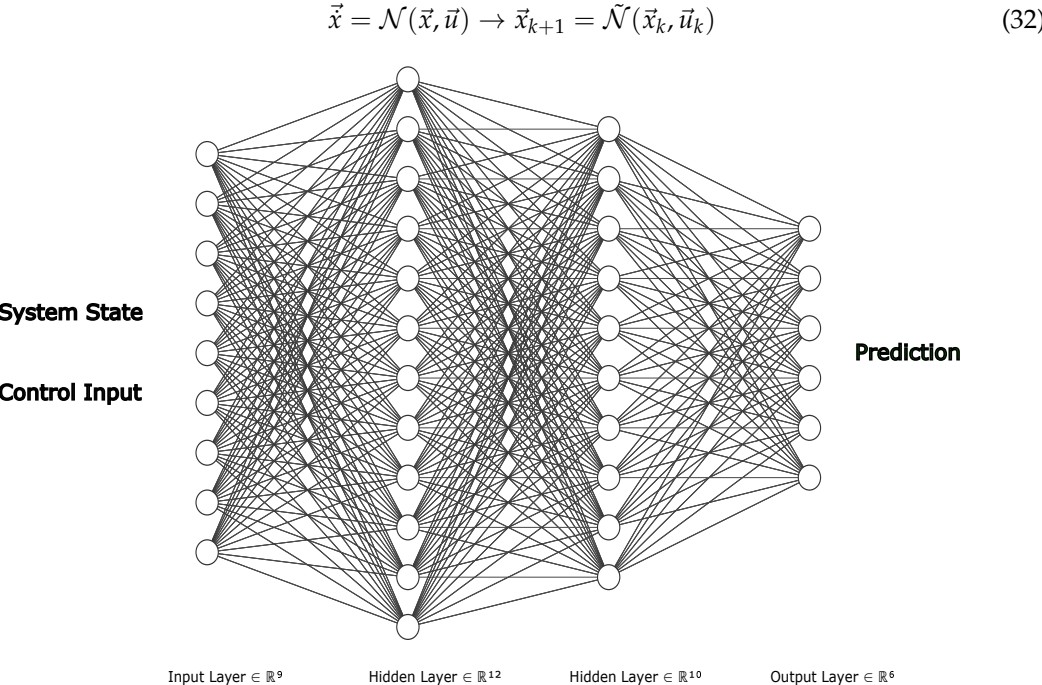

Input Layer $\in \mathbb{R}^9$     Hidden Layer $\in \mathbb{R}^{12}$     Hidden Layer $\in \mathbb{R}^{10}$     Output Layer $\in \mathbb{R}^6$

**Figure 17.** Dynamical reconstruction as a neural network model.

The method relies on the *universal approximation theorem*, since it is based on the assumption that there exists an ANN that approximates the dynamical function with a predefined approximation error. The training set is simply composed of input–output pairs, where the input is a stacked vector of system states and control vectors.

The result is that the full dynamics is encapsulated into a neural network model that can be used to generate prediction of future states [16,54]. The dynamics reconstruction

based solely on artificial neural networks largely benefits by the employment of recurrent neural networks, rather than simpler feedforward networks. Indeed, literature, although rather poor, employ Recurrent Neural Networks to perform the task [25,27,54]. The recurrent architecture owns an inherently more complex structure but maintains a brief evaluation time make it a suitable architecture for on-board applications. As mentioned, Recurrent Networks have the capability of handling time-series data efficiently because the connections between neurons form a directed graph, which allows an internal state memory. This enables the network to exhibit temporal dynamic behaviors. When dealing with dynamics identification, it is crucial to exploit the temporal evolution of the states; hence, RNN shows superior performances with respect to MLP. In [16], the different propagation of the feedforward (MLP) and recurrent neural networks, initialized equally and trained in the same scenario, are compared. The prediction position and velocity accuracy, compared with the analytical nonlinear J2-perturbed model, demonstrates the superior performance in dynamics reconstruction by employing recurrent architectures.

The methodology reported so far relied on FFNN or RNN that are based on the function approximation of the state-space representation of the dynamics. They do not explicitly reconstruct the definition of the Markov problem. An alternative is to use a predictive autoencoder (AEs), a modified version of traditional autoencoders, which is able to learn a nonlinear model in a state-space representation from a dataset comprising input–output tuples [40].

A tentative qualitative summary of the most used ANN architecture for system identification is reported in Table 6. The learning time represents the effectiveness of the incremental learning performed on-board; hence, it is estimated with reference to the characteristic time of the motion, for instance, the orbital period.

**Table 6.** Summary of typical and most successful neural networks for fully neural dynamics learning.

| Application | Type of ANN | Accuracy | Learning Time | Ref. |
|---|---|---|---|---|
| Fully-Neural | MLP | Low | Fast | [27,55] |
| | LRNN | High | Slow | [27] |
| Dynamics Learning | NARX | High | Slow | [27,56,57] |
| | AE | Medium | Fast | [40,58] |

### 4.1.2. Dynamical Uncertainties and Disturbance Reconstruction

The second method uses the capability of the artificial neural networks to approximate an unknown function. In particular, it is wise to exploit all analytical knowledge we may have of the environment. Nevertheless, most of the time, the analytical models encompass linearization and do not model perturbations, either because they are analytically complex or are simply unknown. In view of future space missions involving other bodies, such as asteroids that produce gravitational perturbations which are highly uncertain and are subjected to unknown physical influences, the capability of ANNs to estimate the states of the satellites can also be used to estimate the uncertainties that could be analyzed further while understanding the physical phenomena. To improve the performance of the navigation or the control, neural networks are employed to compensate for the effects of the modeling error, external disturbance and nonlinearities. Most of the researchers focused their efforts on coupling the neural network with a controller to take into account modeling errors, as found in [34,35,38,59]. In this field, the ANN is used to compensate for external disturbance [35], or to compensate for model inversion [59] or spacecraft uncertain properties [34]. Nevertheless, besides eliminating spacecraft uncertainties used in the controller, the system identification can be used also to enhance disturbance estimation and navigation overall, as in [29,30]. In [29], a radial-basis-function-neural-network-aided adaptive extended Kalman filter (RBFNN-AEKF) for state and disturbance estimation was developed. The neural network estimates the unmodeled terms which are fed into the EKF as an additional term to the state and covariance prediction step. The approach proposed by Harl et al. [30], the modified state observer, allows for the estimation of uncertainties

in nonlinear dynamics, and in addition, providing estimates of the system states. The observer structure contains neural networks whose outputs are the uncertainties in the system. In brief, the methods can be summarized into three key-points:

- The ANN is used to learn and output an estimate of the disturbance or mismodeled terms that is used in the guidance, navigation and control to deliver a better state, disturbance or error estimate.
- The ANN learning is fully performed incrementally online. This means that no prior knowledge or learning has to be performed beforehand. This dramatically increases the flexibility of the approach.
- The ANN learning does not replace the GNC system, but it rather enhances it and makes it more robust.

Regardless of the application, the goal of this dynamics reconstruction approach is to reconstruct a perturbation term, whatever the source is. It is remarkable that one could derive an underlying architecture that is common to all the mentioned researches. The idea is to have an observer and a supervised learning method that makes use of the Lyapunov stability theorem to guarantee convergence of the estimation. In general, the simplest form of each algorithm can be developed as follows. Let us assume the actual system dynamics are described by the following set of non-linear differential equations:

$$\dot{\vec{x}} = f(\vec{x}) + d_{ext} \tag{33}$$

where $d$ is the external disturbance term. The actual system dynamics can be rewritten as:

$$\dot{\vec{x}} = A \cdot \vec{x} + d(\vec{x}) \tag{34}$$

where the term $d$ captures all the non-linearities together with the disturbances external to the system, namely, $d = f(\vec{x}) - A + d_{ext}$. The state observer can be constructed as follows:

$$\dot{\hat{\vec{x}}} = A \cdot \hat{\vec{x}} + \hat{d}(\hat{\vec{x}}) + K_h(\vec{x} - \hat{\vec{x}}) \tag{35}$$

where $\hat{d}$ is estimated using the ANN system that we are deploying and $K_h$ is the observer gain matrix. The observer error dynamics can be derived as:

$$\vec{e} = \vec{x} - \hat{\vec{x}} \tag{36}$$

$$\dot{\vec{e}} = \dot{\vec{x}} - \dot{\hat{\vec{x}}} = d(\vec{x}) - \hat{d}(\hat{\vec{x}}) - K_h(\vec{x} - \hat{\vec{x}}) \tag{37}$$

The neural network learning, meaning the weights update rule, generally relies on the observer error dynamics [29,30,59], targeting convergence and stability of the weights matrix $\hat{W}$ and the error $e$. Let us assume that we want to use radial-basis-function neural network: when invoking the universal approximation theorem for neural networks, we can assume there exists an ideal approximation of the disturbance term $d$:

$$d(\vec{x}) = W^T \Phi(\vec{x}) + \epsilon \tag{38}$$

where $\epsilon$ is a bounded arbitrary approximation error. Consequently, the error in estimation can be written as:

$$d(\vec{x}) - \hat{d}(\hat{\vec{x}}) = W^T \Phi(\vec{x}) + \epsilon - \hat{W}^T \Phi(\hat{\vec{x}}) \tag{39}$$

By adding and subtracting the term $W \cdot \Phi(\hat{\vec{x}})$ and performing few mathematical manipulations, Equation (39) can be expressed as:

$$\tilde{d} = \tilde{W}^T \Phi(\hat{\vec{x}}) + \epsilon' \tag{40}$$

where $\tilde{W} = W - \hat{W}$ and the bounded term $\epsilon' = \epsilon + W \cdot [\Phi(\vec{x}) - \Phi(\hat{\vec{x}})]$. The aim of the learning rule is to drive the dynamics error to zero, and also to force the weights to converge to the ideal ones. Namely:

$$\vec{e} \to \vec{0}, \ \tilde{W} \to [\vec{0}]$$

If we reason for a single channel, i.e., the single state approach, by introducing $W_i = [w_{1i} \ w_{2i} \ ... \ w_{mi}]^T$, $i$ = 1:6, the following Lyapunov function can be constructed to derive the stable learning rule:

$$V_i = \frac{1}{2}\tilde{W}_i^T\tilde{W}_i + \frac{1}{2}e_i^2 \tag{41}$$

By recalling Equations (37) and (40), the derivative of the Lyapunov function can be written as:

$$\dot{V}_i = \tilde{W}_i^T(\dot{\tilde{W}}_i + e_i\Phi(\hat{\vec{x}})) + e_i\epsilon' - K_he_i^2 \tag{42}$$

In order to fulfill the hypothesis of the Lyapunov stability theorem, the weights update rule is derived to drive the derivative in Equation (42) < 0. Recalling from Equation (40) that $\dot{\tilde{W}} = -\dot{\hat{W}}$, the expression for a single-state weight update rule:

$$\dot{\hat{W}}_i = e_i\Phi(\hat{\vec{x}}) \tag{43}$$

In compact form, we can express the weights matrix update rule:

$$\dot{\hat{W}} = \Phi(\hat{\vec{x}})\vec{e}^T \tag{44}$$

The reader may want to modify the presented core algorithm into a more sophisticated architecture. Nevertheless, the literature currently adopts this as a consolidated approach. A tentative qualitative summary of the most used ANN architecture for a disturbance dynamics term reconstruction is reported in Table 7. As previously mentioned, the learning time represents the effectiveness of the incremental learning performed on-board; hence, it is estimated with reference to the characteristic time of the motion, for instance, the orbital period.

**Table 7.** Summary of typical and most successful neural networks for dynamics uncertainty learning.

| Application | Type of ANN | Accuracy | Learning Time | Ref. |
|---|---|---|---|---|
| Uncertainties and | MLP | Medium | Medium | [34,35,38,59] |
| disturbance estimation | RBFNN | High | Fast | [29,30,60] |

4.1.3. System Identification through Reconstruction of Parameters

The third method has been developed under the framework of parameter reconstruction. Basically, the artificial neural network is employed to refine the uncertain parameters of a given dynamical model. This method is particularly suitable when the uncertain environment influences primal system constants (e.g., inertia parameters, spherical harmonics and drag coefficients). Differently from the disturbance approximation, the analytical framework is here known, and only characteristic values of system parameters are reconstructed. Moreover, given the physical knowledge of the parameters to be reconstructed, the method has a very promising *scientific* outcome. For instance, the gravity expansion of asteroids and planets can be approximated online while flying, delivering a rough shape reconstruction of the body. Nevertheless, many applications overlap with the disturbance reconstruction approach, in which the disturbance function to approximate is a constant parameter, as assumed in many estimation algorithms, such as the well-established Kalman Filter. For this reason, many researchers used MLP neural networks to carry out the task. For instance, Chu et al. [36] proposed a deep network MLP to estimate inertia parameters. The angular rates and control torques of combined spacecraft are set as the input of a deep

neural network model, and conversely, the inertia tensor is then set as the output. Training the MLP model refers to the process of extracting a higher abstract feature, i.e., the inertia tensor [36,37]. Another approach to solve the parametric reconstruction is to use a recurrent neural network. In particular, Hopfield neural networks are investigated in [44,45,61]. The core of the algorithm is the following: if the model is reformulated into linear-in-parameter form, the identification problem can be reformulated as an optimization problem:

$$\vec{y} = A(\vec{r}) \cdot \vec{C} \tag{45}$$

In particular, when defining a general *prediction error* $\vec{e} = \vec{y} - A \cdot \vec{C}^*$, the resulting combinatorial optimization problem is [44]:

$$\min_{\vec{C}} \left\{ \sup_t \left( \frac{1}{2} \vec{e}^T \cdot \vec{e} \right) \right\} \tag{46}$$

where $\vec{y}$ typically correspond to measurements, A is the linear-in-parameter matrix and $\vec{C}^*$ is the estimated parameter vector [44]. Other examples entail the usage of BPANN, another name for MLP, for the Earth or other bodies gravity field approximation, as presented in [62,63]. A tentative qualitative summary of the most used ANN architecture for parametric dynamics reconstruction is reported in Table 8.

**Table 8.** Summary of typical and most successful neural networks for parametric learning. The learning times for the MLP applications are not available.

| Application | Type of ANN | Accuracy | Learning Time | Ref. |
|---|---|---|---|---|
| System Identification | MLP | High | - | [36,37,62,63] |
| | HNN | High | Fast | [44,45,61] |

*4.2. Convolutional Neural Networks for Vision-Based Navigation*

Convolutional neural networks are the best candidates with which to process visual data. For this reason, CNNs have quickly become of great interest to vision-based navigation designers. In particular, CNNs have been employed in end-to-end or hybrid approaches for spacecraft navigation, mainly in two scenarios: proximity operations during a close approach with uncooperative targets and planetary or asteroid landing. The algorithm architecture typically relies on a CNN that either segments or classifies images, whose output is fed to the navigation system, composed of estimation and sensor fusion algorithms.

4.2.1. CNN for Pose Estimation

CNNs have been recently becoming a promising solution for the pose estimation and initialization of target spacecraft. Most of the algorithms used in spacecraft GNC are hybrid approaches and make use of artificial neural networks only partially. Nevertheless, end-to-end methods are currently being explored. A targeted review of CNNs for pose estimation is reported in [64]. Generally, in a CNN-based method, a pre-training is necessary to develop a model capable of performing regression or classification to estimate the pose. The monocular image is then fed to such a model online to retrieve the pose. Depending on the selected architecture adopted to solve for the relative pose, these methods can either rely on a wire-frame 3D model of the target spacecraft or solely on the 2D images used in the training, and hence they can either be referred to as non-model based or model-based [64]. One solution to the problem of satellite pose estimation was provided by [65]. It is a typical approach for model-based hybrid architecture, exploiting object detection networks and requiring 3D models of the spacecraft. It consists of a monocular pose estimation technique composed of different steps:

1. An object detection network is used to identify a bounding box surrounding the target spacecraft. Typical CNN architectures are HRNet and Faster RCNN.
2. A second regressive network is used to predict the position of the landmark features utilized during training.
3. A traditional PnP problem was solved using 2D–3D correspondences to retrieve the camera pose.

Another approach is the end-to-end pose estimation architecture presented in [66]. The approach entails a feature extraction method using a pre-trained CNN (modified ResNet 50) backbone. CNN features are fed into a sub-sampling convolution for compression. The generated intermediate feature vectors are then used as input to two separate branches: the first branch uses a shallow MLP that regresses the 3D locations of the features. The second branch uses probabilistic orientation soft classification to generate an orientation estimate. Sharma et al. [67] proposed a three-branch architecture. The first branch of the CNN bootstraps a state-of-the-art object detection algorithm to detect a 2D bounding box around the target spacecraft in the input image. The region inside the 2D bounding box is then used by the other two branches of the CNN to determine the relative attitude by initially classifying the input region into discrete coarse attitude labels before regressing to a finer estimate. The SPN method then uses a novel Gauss–Newton algorithm to estimate the relative position by using the constraints imposed by the detected 2D bounding box and the estimated relative attitude. Recently, the approach presented in [68] replicates the three-module structure entailing the object detection, keypoint regression and single image pose estimation in three different neural networks. The first module uses an object detection network to generate a 2D bounding box. Such task is executed by a CNN YOLOv3 with a MobileNetv2 backbone, which are well-established CNN architectures. The bounding box output is used in the second module to extract a region of interest from the input image. The cropped image is fed to a keypoint regression network (KRN) which regresses locations of known surface keypoints of the target. Eventually, a traditional PnP solver is used to retrieve the single-image pose estimation. Given the usage of PnP algorithms, the estimated 2D keypoints require the corresponding 3D locations, meaning that a 3D model of the spacecraft is needed. Nevertheless, a potential solution to the model unavailability is the 3D keypoints recovery preparatory step. Finally, Pasqualetto et al. [69] investigated the potentials of using a hourglass neural network to extract the corners of a target spacecraft prior to the pose estimation. In this method, the output of neural network is a set of so called heatmaps around the features used in the offline training. The coordinates of each heatmap's peak intensity characterize the predicted feature location; the intensity indicates the confidence of locating the corresponding keypoint at this position.

### 4.2.2. CNNs for Planetary and Asteroid Landing

The use of AI in scenarios for the moon or planetary landing is still at an early stage, and few works exist in the literature concerning image-based navigation with AI. The presented solutions are all based on a supervised learning approach. Images with different lighting and surface viewing conditions are used for training. One idea is to substitute classical IP algorithms, providing a first estimate of the lander state (e.g., pose or altitude and position), which can be later refined by means of a navigation filter. Similar approaches have been implemented in the robotics field, where by pre-training on a large dataset, the methods estimate the absolute camera pose in a scene. In a landing scenario, the knowledge of the landing area can be exploited, if available. Therefore, the convolutional neural network can be trained with an appropriate dataset of synthetic images of the landing area at different relative poses and in different illumination conditions. The CNN is used to extract features that are then passed to a fully connected layer, which performs a regression and directly outputs the absolute camera pose. The regression task can be executed by an LSTM. The CNN-LSTM has proven excellent performance and is very well developed for image processing and model prediction. The use of a recurrent network brings the advantage of also retrieving time-series information. This can allow also estimating the

velocity of the lander. According to an extensive review of the applications, one can make a general distinction between two macro-methods:

- Hybrid approaches: they utilize CNNs for processing images, extracting features and classifying or regressing the state at the initial condition, but they are always coupled with traditional image processing or a navigation algorithm (e.g., PnP and feature tracking).
- End-to-end approaches: they are developed to complete the whole visual odometry pipeline, from the image input to the state estimate output.

As an interesting example, although not directly applied to space systems, the technique presented in [70] relies on end-to-end learning for estimating the pose of a UAV during landing. In particular, the global position and orientation of the robot are the final output of the AI architecture. The AI system processes two kinds of inputs: images and measurements from an IMU. The architecture comprises a CNN that takes as input streams of images and acts as a feature extractor. Such a CNN is built starting from ResNet18, pre-trained on the ImageNet dataset. An LSTM processes the IMU measurements, which are available at a higher frequency than images. An intermediate fully connected layer fuses the inertial and visual features coming from the CNN and the LSTM. Then, such vector is passed to the core LSTM, along with the previous hidden state, allowing one to model the dynamics and connections between sequences of features. Finally, a fully connected layer maps the feature to the desired pose output. Similarly, the architecture proposed by Furfaro et al. [25] comprises a CNN and an LSTM. The final output of the AI system is a thrust profile to control the spacecraft landing. The CNN's input consists of three subsequent static images. This choice is motivated by the need of retrieving some dynamical information [25]. The whole visual odometry pipeline has been learned completely in the work by Wang [71]. The approach proposed in [71] exploits a deep learning system based on a monocular visual odometry (VO) algorithm to estimate poses from raw RGB images. Since it is trained and deployed in an end-to-end manner, it infers poses directly from a sequence of raw RGB images without adopting any module in the conventional VO pipeline. The AI system comprises the convolutional neural network that automatically learns effective feature representation for the visual odometry problem, but also a recurrent network, which implicitly models sequential dynamics and relations. The final output is the absolute pose of the vehicle. This architecture differs from the one presented in [70], because here two consecutive frames are stacked together and only images are considered as inputs. A hybrid approach specifically developed for lunar landing was presented by [72] and re-adapted by [14,73,74]. The approach is based on the work by Silburt et al. [75]: a deep learning approach was used to identify lunar craters; in particular, a Unet-CNN, shown in Figure 18, was used for input images' segmentation, as shown in Figure 19. Some traditional navigation strategies are based on lunar crater matching; therefore, an AI method was investigated as part of a hybrid approach, as in [72,76,77], where a RANSAC-based nearest neighbor algorithm was used for matching the detected craters to database ones. The advantage of such a hybrid approach is to combine a crater detection method that is robust to illumination conditions and the reliability of a traditional pose estimation pipeline. An example of an input image and an output mask is shown in Figure 18. This is a powerful technique for absolute navigation where database objects can be used for learning. The state estimation requires a navigation filter or feature postprocessing, such as computation of an essential matrix or retrieval of relative vectors, to complete the navigation pipeline. A qualitative summary of the most promising methods is reported in Table 9.

A very interesting application of an ANN to planetary landing is the autonomous collision avoidance presented by Lunghi et al. [78]. An MLP is fed with an image of the landing area. The neural system outputs a hazard map, which is then exploited to select the best target, in terms of safety, guidance constraints and scientific interest. The ANN's generalization properties allow the system to correctly operate also in conditions not explicitly considered during calibration.

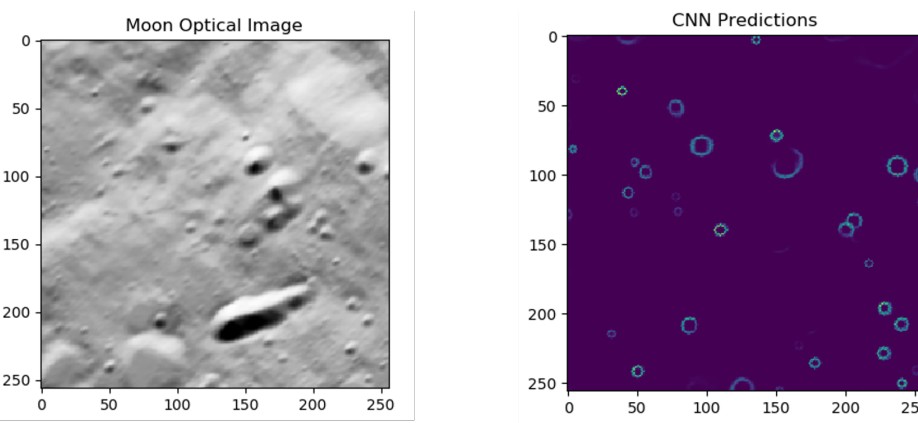

**Figure 18.** Input image and output mask from the CNN [14].

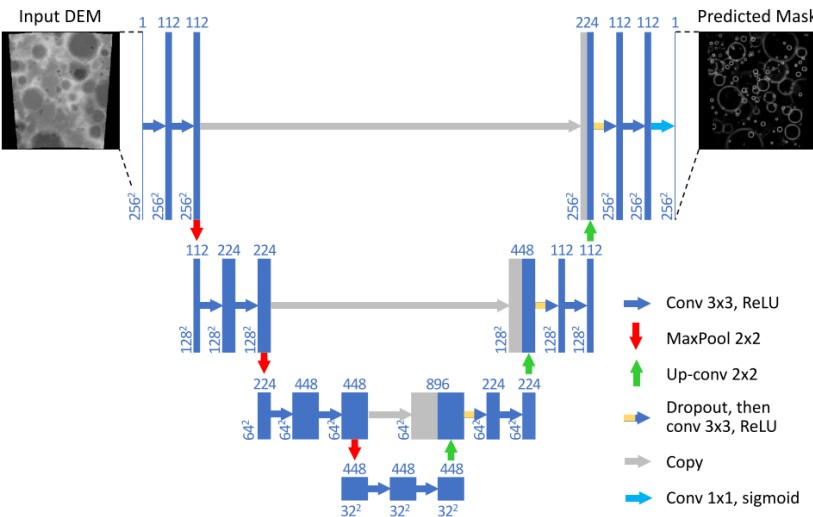

**Figure 19.** Convolutional neural network (CNN) architecture, based on Unet [75,79]. Boxes represent cross-sections of sets of square feature maps.

**Table 9.** Summary of neural-aided methods for optical navigation for planetary and lunar landing. IP stands for image processing and represents the necessity for an hybrid method to be implemented (e.g., matching and PnP).

| Application | Type | Accuracy | Training Needs | Robustness | Adaptability |
|---|---|---|---|---|---|
| Optical | CNN + IP | High | Medium | High | Medium |
| Navigation | CNN + LSTM | Medium | High | High | High |

*4.3. Reinforcement Learning and Meta-Reinforcement Learning for Adaptive Guidance and Control*

4.3.1. Reinforcement Learning

An active research field is to use reinforcement learning and meta reinforcement learning to create an adaptive guidance and control system. In particular, deep reinforcement learning has been used to generate autonomous guidance and control during proximity operations and landing trajectories [23,24,80]. Reinforcement learning has been used to generate autonomous trajectory planning for different scopes. Pesce et al. [81], Piccinin et al. [82,83] and Chan et al. [84] analyzed the autonomous mapping of asteroids using DQN and neural fitted Q as value-based methods. Federici et al. [85] proposed an actor–critic proximal policy optimization framework for real-time optimal spacecraft guidance during terminal rendezvous maneuvers, in the presence of both operational constraints and stochastic effects, such as inaccurate knowledge of the initial spacecraft state and the presence of random in-flight disturbances. Brandonisio et al. [24] proposed a

guidance and control law to perform the inspection of an uncooperative spacecraft. In [24], a DQN and advantage actor–critic (A2C) methods. State–action value functions are approximated using artificial neural networks (ANN); in particular, simple MLPs are employed. A discrete action space is maintained, whereas the state space is continuous. Transfer learning (TL) is also applied to facilitate training on more complex tests. One of the transfer learning techniques consists of pre-training the RL agent on a simpler task before training on the main task. In the paper, the various tasks are represented by increasing complexities of the reward models. Many researchers claim superior performance by policy-based methods. Among those, proximal policy optimization (PPO) and derivatives are some of the most adopted schemes [17,18,85]. In order to improve the stability and robustness of the agent in different scenario conditions, a formulation of PPO exploiting recurrent neural networks (RNN) is commonly exploited. As anticipated in Section 3, the capability of recurrent layers to store past states' information may strongly affect the agent-safe trajectory planning and getting to the mission goals faster. In addition, the work on [17] claims that training an RNN is beneficial to refining the agent's environmental conditions sensitivity, which works in favor of the agent's robustness, regardless of the specific operational environment. The work in [22] proposes a guidance strategy for spacecraft proximity tracking operations leveraging deep reinforcement learning. In [22], the distributed distributional deep deterministic policy gradient (D4PG) algorithm was used. Such algorithm operates in continuous state and action spaces, and it has a deterministic output. The D4PG algorithm is an extension of the actor–critic algorithm.

### 4.3.2. Meta Reinforcement Learning

The robustness to uncertain spacecraft model and environments is a crucial topic on the development of autonomous systems. In particular, this aspect is very challenging when it comes to the the reinforcement learning methods. Neural networks learn well within the training distributions, but they generally fail when performing extrapolation outside the training distributions [21]. This may pose a risk of instability of the guidance law when the spacecraft experiences states that are outside the training distribution envelope. Several researchers [19,21] claim that sampling inefficiency is another weakness of traditional reinforcement learning: a large amount of experience is needed to learn even the simple tasks. In [86] was proposed a reinforcement learning framework to control a spacecraft around a small celestial body with an unknown gravity field. In particular, the hovering task was investigated. The authors performed a direct policy search (DPS) with a genetic algorithm to obtain controller policies with high utility. The policy architecture used was a simple MLP.

Recent advancements in meta reinforcement learning aimed at addressing these weaknesses of the traditional framework. Meta reinforcement learning trains the policy agent on a distribution of environments, or Markov decision processes (MDP). This forces the agent to experience and learn multiple different situations. Consequently, the system tends to converge faster to quasi-optimal solutions. Recent works claim superior performance of meta-RL with respect to classical RL when uncertain environments and actuator failures are considered [18,20,21]. Gaudet et al. [18,20] developed a guidance law based on meta-RL which is able to perform a six degrees-of-freedom Mars landing. Moreover, in [20], meta reinforcment learning was used to create a guidance law for hovering on irregularly shaped asteroids using LIDAR sensor data. In certain works, the different environments are called tasks: the tasks can be thought of as ensembles of potential situations, nominal and non-nominal, one can expect the agent to experience. For instance, in the application of meta-RL for planetary and asteroid landing [18], the tasks range from landing with engine failures to large mass variations or highly-corrupted navigation and unknown dynamics.

In [87], a meta reinforcement learning framework is employed for relative trajectory planning between spacecraft. The training trajectories are divided as sub-training samples and fake testing samples. The meta-reinforced agent is trained by alternating training and testing phases. To this end, the gradient information of the meta learner is obtained through

a combination of the results on the training subsets and the performance on the fake testing samples. The authors of [87] claimed that this approach forces the agent to explicitly take the potential testing performance into consideration. In this way, the overfitting phenomenon, potentially arising using few training trajectories, is reduced. In most of the studies works, as for traditional reinforcement learning applications, the meta reinforcement learning policy is optimized using proximal policy optimization (PPO) [18,21] with both the policy and value functions implementing recurrent layers in their networks. Using recurrent neural networks results in creating agents that can adapt more easily to uncertain environments, yielding a much more robust guidance policy compared with classical reinforcement learning [19]. If we examine a particular scenario, such as planetary landing, by recalling what was described in Section 3, it is easier to understand how recurrent layers result in an adaptive agent. During the training to generate an autonomous landing agent, the next observation depends not only on the state and action, but also on the ground truth agent mass and any external forces acting on the agent at each step [18]. Consequently, during training, the recurrent layers force their hidden states to evolve differently depending on the observations acquired from the environment during the trajectory, which are governed by the actions output by the policy. Specifically, the trained policy's hidden state captures unobserved information, such as external forces or agent mass, that is useful in minimizing the cost function [18]. Obviously, this holds for any scenario one could be interested in, even those not related to space operations.

### 4.4. Image Dataset Availability and Validation

The successful deployment of ML and AI techinques requires a large amount of data to properly train, validate and test the applications. In particular, this review reported several examples where image-based algorithms use optical measurements to synthesize the navigation, generate the guidance or even to control the spacecraft directly. One major challenge in space AI-based algorithm development is the lack of representative data. This obviously comes from the limited availability of the space environment, with respect to other scenarios, such as autonomous terrestrial driving, for instance. Hence, it is crucial to focus the research on the generation and validation of non-space images that can be used as representative for orbital applications.

According to the state of art, several laboratory setups exist to recreate rendezvous approaches around a mockup of a target space object (i.e., asteroid, moon surface or spacecraft) with a monocular camera, i.e., the Testbed for Rendezvous and Optical Navigation (TRON) at Stanford University [88], the GNC Rendezvous, Approach and Landing Simulator (GRALS) at the European Space Research and Technology Centre (ESTEC) [89], the European Proximity Operations Simulator (EPOS) at the German Aerospace Agency (DLR) [90], the Platform-art facility at GMV [91] and the ARGOS facility at Politecnico di Milano [92,93].

In this context, a fundamental challenge arises from the need to bridge the quality gap between the synthetic renderings and the laboratory-representative images. If a synthetic dataset used to train the adopted CNN fails at representing the textures of the target mockup and the specific illumination in the laboratory setup, the performance on laboratory-generated images will in fact result in inaccurate pose estimates. To overcome this, recent works addressed the impact of augmented synthetic datasets on CNN performance in either laboratory-generated or space-based imagery [88,93–96]. These augmented datasets were built on a backbone of purely synthetic images of the target by adding noise; randomized and real Earth backgrounds; and randomized textures of the target model.

By condensing the literature outcomes, we can point at four indications to evaluate the representativeness of the synthetic images with respect to laboratory ones (or real, if available) [93]:

1.  **Image histogram**. The histogram's information is a low-level information, which gives a good representation of the image content. Such a method has been already used to evaluate images quality for testing of space navigation algorithms [88].

2. **Shadow index**. The synthetic and laboratory images are thresholded to identify shadows. The value of the threshold is identified automatically using the Otsu algorithm. The Otsu method is a deterministic and automatic way to discriminate shadowy and illuminated target parts. Then, the two resulting binary images are subtracted to obtain a shadow disparity map. The accuracy of the shadow representation, which can be considered as representative of the accuracy of the general shape of the sample, is evaluated by a scalar shadow index ($J_s$), defined as:

$$J_s = 1 - \frac{D_s}{S_{real}} \tag{47}$$

where $D_s$ is the sum of the disparity map and $S_{real}$ is the sum of the pixels classified as shadow in the real image. $J_s$ expresses the fraction of pixels in shadows correctly reproduced in the synthetic model.

3. **Contrast index**. A second index is then identified. For both images, real and synthetic, an illumination ratio RI is identified as:

$$R_I = \frac{I_L}{I_S} \tag{48}$$

where $I_L$ is the mean intensity of the pixel classified as in light, and $I_S$ is the mean intensity of the pixel classified as in shadow. Then, the contrast index $J_c$ is defined as:

$$J_c = \frac{R_{Ireal}}{R_{Irend}} \tag{49}$$

4. **Feature quality index**. Typical navigation algorithms rely on feature extraction steps; thus, a comparison among real and synthetic images is considered a good indication of the similarity of behavior among the two. The feature quality index (FQI) indicates the similarity of features extracted in two corresponding frames (a real and a synthetic one), and it is defined as:

$$FQI = 1 - \frac{\mu(H_d)}{H_{d,max}} \tag{50}$$

where $H_d$ is the Hamming distance between two corresponding features descriptors and $H_{d,max}$ is the maximum possible hamming distance. The mean value $\mu(H_d)$ is computed on a user-defined set of corresponding matched features.

The higher the indexes, the better the image is represented. The requirements to satisfy are user defined and still represent an arbitrary variable in the process. For instance, in [93] it is required $J_c > 0.90$, $J_s > 0.75$ and $FQI > 0.80$.

*4.5. Technical Challenges for AI-Based Algorithms' Deployment*

The technical challenges for the consistent deployment of AI-based solutions to actual spacecraft are heterogeneous. This section provides a very brief outline of the main topics where research is active to pursue the above-mentioned objective.

- *Data Availability*: A large amount of data is required to generate effective AI-based algorithms. One critical problem is creating representative data, especially images, in synthetic or laboratory environments, as discussed in Section 4.4.
- *Model Compression*: High-performance AI solutions, based on DNN, often require very large models to be deployed. This negatively affects the following aspects:
  - Storage capacity: A DNN model can achieve significant accuracy when it uses a large number of parameters, which requires considerable storage.
  - Computational requirements: A large number of floating point operations (FLOPs) involved in the DNN operation can exceed the limited computational capacity of the dedicated hardware.

– Execution time: A large DNN model requires a long time for both training and inference. This could potentially jeopardize its real-time inference performance.

Optimization and model compression techniques are currently being investigated, such as pruning and weight sharing, which help also in terms of energy consumption.

- *Validation*: Given the difficulties in making the AI-based algorithms analytically tractable, it is mandatory to establish a consolidated pipeline to validate the models. Monte Carlo approaches may be the most appropriate solutions to characterizing the behavior, also outside of the training datasets, where inappropriate responses may yield dramatic outcomes.
- *Dedicated Hardware*: A lot of effort is focused on building and testing dedicated hardware, particularly tailored to execute AI-based models, optimized for inference.

## 5. Conclusions

This paper presented an overview of the applications of machine learning, deep learning and artificial neural networks in the spacecraft guidance, navigation and control domain. In particular, a brief outline of the theoretical foundations of the Artificial-Intelligence-based methods has been presented, in order to provide the reader a tailored introduction to the novel approaches. The goal of the paper was to highlight the concepts of Artificial Intelligence that are spreading in the space community and to underline what are the constraints and limitations of these methods in the challenging space environment. A thorough review of the most employed neural network architectures has been presented, along with the working principle behind each of them. The peculiarity of each artificial neural network has been stressed and linked to specific applications in the research domain. One of the most interesting applications in the spacecraft dynamics, guidance, navigation and control domain involves the usage of sensed data to retrieve temporal structures, approximate disturbances, encapsulate the dynamical behavior or perform parametric system identification. Typically, recurrent neural networks exhibit superior performance in approximating temporal series, at the cost of high training complexity. The wide pool of convolutional neural networks are employed to process images coming from optical sensors, meaning that the most promising application is to couple neural-based methods and image processing techniques to carry out the task of optical navigation. Indeed, two applications were reviewed which report the state-of-the-art in CNN-based methods for pose estimation and planetary landing. Additionally, the paper introduced the vast domain of deep reinforcement learning, analyzing its applications to autonomous guidance and control in multiple scenarios, such as planetary landing and proximity operations. Moreover, different strategies to increase algorithms' robustness were reviewed and described, such as using transfer-learning and meta-reinforcement-learning approaches. However, there are some unresolved challenges to applying AI algorithms to GNC systems, for instance, inadequate data sets for training, the theoretical understanding and modeling of the behavior of any AI system, the generalization of the learned features to different scenarios and the validation. Moreover, we point out two additional shortcomings of the current status of ML and AI in view of spacecraft GNC deployment:

- The trade-off between adaptivity and robustness in the design of the GNC system. On the one hand, we are trying to design machine learning systems that evolve continuously by learning via interaction with the dynamical and physical environment. On the other hand, we should pursue optimized solutions that are robust, explainable and secure.
- The AI and ML algorithms borrowed from data science often lack efficiency, robustness and interpretation, being purely data-driven approaches. The foundation of classical GNC theory instead lies in the mapping of physics into the model-based design concept.

In conclusion, we aimed to developing a useful review and tutorial for professionals in the space sector that specifically want to adopt deep learning and artificial neural networks

or desire to be updated on the most relevant and pertinent concepts of Artificial Intelligence for space applications.

**Author Contributions:** Conceptualization, S.S.; methodology, S.S.; software, S.S.; validation, S.S.; formal analysis, S.S.; investigation, S.S.; resources, S.S.; data curation, S.S.; writing—original draft preparation, S.S.; writing—review and editing, S.S.; visualization, S.S.; supervision, M.L.; project administration, M.L.; funding acquisition, M.L. All authors have read and agreed to the published version of the manuscript.

**Funding:** This research received no external funding.

**Institutional Review Board Statement:** Not applicable.

**Informed Consent Statement:** Informed consent was obtained from all subjects involved in the study.

**Data Availability Statement:** Data sharing not applicable

**Conflicts of Interest:** The authors declare no conflict of interest.

## Abbreviations

The following abbreviations are used in this manuscript:

| | |
|---|---|
| A2C | Advantage Actor Critic |
| AE | Autoencoders |
| AEKF | Adaptive Extended Kalman Filter |
| AI | Artificial Intelligence |
| ANN | Artificial Neural Network |
| BPANN | Back Propagation Artificial Neural Network |
| CNN | Convolutional Neural Network |
| D4PG | Distributed Distributional Deep Deterministic Policy Gradient |
| DCG | Directed Cyclic Graph |
| DL | Deep Learning |
| DPS | Direct Policy Search |
| DQN | Deep Q-Network |
| FFNN | Feed-Forward Neural Networks |
| GNC | Guidance, Navigation & Control |
| GRU | Gated-Recurrent Unit |
| HNN | Hopfield Neural Network |
| IF | Integrate and Fire |
| IMU | Inertial Measurement Unit |
| KRN | Keypoint Regression Network |
| LIF | Leaky-Integrate and Fire |
| LRNN | Layer-Recurrent Neural Network |
| LSTM | Long-Short Term Memory network |
| MDP | Markov Decision Process |
| ML | Machine Learning |
| MLP | Multi-Layer Perceptron |
| NARX | Nonlinear Autoregressive Exogenous Model |
| PnP | Perspective n-Points |
| PPO | Proximal Policy Optimization |
| RBF | Radial-Basis Function |
| RBFNN | Radial-Basis Function Neural Network |
| RL | Reinforcement Learning |
| RNN | Recurrent Neural Network |
| SNN | Spiking Neural Network |
| TL | Transfer Learning |
| VO | Visual Odometry |

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
