# Peer review of "Deep Learning and Artificial Neural Networks for Spacecraft Dynamics, Navigation and Control"

_drones, doi:10.3390/drones6100270_

Round 1
Reviewer 1 Report
Thank you for delivering a nice manuscript related to the integration of deep learning and artificial neural networks for spacecraft dynamics, navigation and control. Both Abstracts and Conclusions are easy for readers to understand this work. The writing and presentation quality of this work is nice. This overview presents the end-to-end deep learning frameworks for spacecraft guidance, navigation and control together with the hybrid methods in which the neural techniques are coupled with traditional algorithms to enhance their performance level. I don’t have any specific comments as the authors have nicely presented this work and clearly explained their research contributions. A few suggestions are
1. Please add an acronym Table to define all abbreviations.
2. It is suggested to provide a more comprehensive discussion in the Introduction section.
3. It will be interesting to add a comparative Table in the Introduction section and compare your work with existing reviews or surveys.
4. It is suggested to provide a precise summary in the start of each section to define contributions in it.
5. Figures are properly labeled. However, the Tables should be more comprehensive so readers can find complete information from Tables.
6. There is a complete lack of discussion and references in Subsections 2.1 and 2.2.
7. I suggest authors must add a comprehensive Table in Section 2 for different learning approaches.
8. Subsections 2.4.4 and 2.4.5 are reported without any reference. Add some relevant references.
9. Again, there is a lack of reference literature in Section 3. Some subsections are discussed without any reference.
10. It will be interesting if the authors can add a few technical challenges in a separate section before the conclusion to further enhance the quality of this study. However, it is not mandatory as the authors have provided enough contributions to this review.
11. I suggest adding a few more references from research works conducted in 2022.
12. The Conclusion must be revised and make it more precise. Also, add a few possible future research directions in this domain.
In my opinion, the overall research contributions are good and satisfying to be published after these minor revisions.
Reviewer 2 Report
First 20 pages are focused on general AI and ANN techniques state of the art, implication and hence comparison by differentiation. Space specific topic is addressed at section 4.
The paper is a very good review of available options and techniques. I suggest acceptance in present form but still good like to make some suggestion, that author can take into consideration either for present paper or for the future.
Did the authors analyse over DNN the different optimization techniques? Pruning, weight sharing, quantization....
One problematic for the space adoption that is not fully addressed is the autonomous concept deployment in embedded processing platforms enduring space environment. Because AI in space domain can be very different if it is to be applicable to ground data processing, to Earth Observation missions or to such a critical scenario as the GNC for Landing that the author is addressing.
Optimization is directly tackling part of this problematic to correlate with growing requirements in terms of memory and processing power, which certainly challenge on-board implementations.
For the specific use-case of planetary/asteroid landing, the topic on CNN/DNNs may also re-inforce analysis on arcihtectures with further explanations, such as commented YoloV3 or U-Net appearing in one of the references.
On other topics, is the author considering any available or to be generated dataset? Data, data, and more data, it is one of the crucial parts on deploying DNN or ML solutions. I suggest including, for a next publication on the topic, dedicated section for optical navigation that might address Image Datasets and AI validation.
Reviewer 3 Report
The paper is well written with a clear emphasis on the contributions.
Contribution # 4 shows the performance evaluation while the whole article doesn't show any performance evaluation.
What is the main contribution as compared with the https://arxiv.org/pdf/2108.08876.pdf?
In section 3 of the article, all the types of neural networks are explained in general, but in contribution, it is shown specifically for the space domain.
As a whole, the article is not correlated as per the contributions mentioned in section 1.
Round 2
Reviewer 3 Report
In the contribution section, the authors claims that the section 4 presents "to evaluate the performance of different neural approaches used in guidance, navigation, and control application".
But in section 4, this is not the performance evaluation?? the authors are required to clearly state the section 4 presents the performance comparison of existing techniques with each other.
Author Response
Dear reviewer, I apologize if I did not understand the comment at first. The contribution bullet point has been changed into "to provide the performance comparison of different neural approaches used in guidance, navigation and control application that exists in literature."
I hope this implements the comment correctly.